6

- Spatially Contrasting CO<sub>2</sub> Dynamics Driven by Green Manure
- 2 Intercropping in Subtropical Tea Plantations

| 4 | Shuo Liu <sup>1,4</sup> | , Zeping Ji | n <sup>1,3</sup> , Ziyi | Chen <sup>1,3</sup> , | Haolin I | Li <sup>1,3</sup> , | Zihan | Fan <sup>3</sup> , | Shaohui | Li <sup>3</sup> | , |
|---|-------------------------|-------------|-------------------------|-----------------------|----------|---------------------|-------|--------------------|---------|-----------------|---|
|---|-------------------------|-------------|-------------------------|-----------------------|----------|---------------------|-------|--------------------|---------|-----------------|---|

- 5 Haiwang Fu<sup>1,4</sup>, Wei He<sup>1</sup>, Kunpeng Zang<sup>1</sup>, Shuangxi Fang<sup>1,5\*</sup>, Peng Yan<sup>2</sup>
- Legiang Carbon Neutral Innovation Institute & Zhejiang International Cooperation Base
- for Science and Technology on Carbon Emission Reduction and Monitoring, Zhejiang
- University of Technology, Hangzhou 310014, China
- <sup>2</sup> Key Laboratory of Tea Quality and Safety Control, Ministry of Agriculture, Tea Research
- Institute, Chinese Academy of Agricultural Sciences, Hangzhou 310008, China
- <sup>3</sup> College of Environment, Zhejiang University of Technology, Hangzhou 310014, China
- <sup>4</sup> Shaoxing Research Institute, Zhejiang University of Technology, Shaoxing 312077, China
- State Key Laboratory of Green Chemical Synthesis and Conversion, Zhejiang University of
- Technology, Hangzhou 310014, China
- Correspondence authors:
- Shuangxi Fang, E-mail: fangsx@zjut.edu.cn

### Abstract:

46

20 Tea plantations are important contributors to greenhouse gas emissions due to intensive 21 fertilization and continuous cultivation. However, the mechanisms by which green 22 manure intercropping regulates soil CO<sub>2</sub> dynamics in these systems remain poorly 23 understood. We investigated how intercropping with Vulpia myuros (SM) and a legume-nonlegume mixture of Lolium perenne and Trifolium repens (HM) influenced 24 25 spatial CO2 flux dynamics compared with a no-intercropping control (CK) from tea rows and inter-row zones in a subtropical tea plantation. Distinct seasonal variations 26 were observed, with CO<sub>2</sub> fluxes peaking in summer and autumn and declining in spring 27 and winter. Average tea-row fluxes were 7.41  $\pm$  0.45, 7.35  $\pm$  0.44, and 8.12  $\pm$  0.46 28 mg·m<sup>-2</sup>·min<sup>-1</sup> under SM, HM, and CK, respectively, indicating emission reductions 29 with intercropping. In contrast, inter-row fluxes were higher under SM (10.83  $\pm$  0.52 30 mg m $^{-2}$  min $^{-1}$ ) and HM (9.77  $\pm$  0.54 mg m $^{-2}$  min $^{-1}$ ) than under CK (9.07  $\pm$  0.44 mg m $^{-2}$ 31 32 min<sup>-1</sup>), demonstrating pronounced spatial contrasts. Diurnal patterns exhibited midday peaks (12:00-14:00), especially in spring and summer, and short-term CO<sub>2</sub> pulses were 33 triggered by field operations such as fertilization and pruning. Notably, HM effectively 34 35 suppressed fertilization-induced CO2 pulses, revealing the mitigation potential of 36 legume-nonlegume mixtures. Green manure increased soil organic carbon (6.4%), 37 lowered soil temperature (4.5%), and enhanced porosity (4.2%), collectively shaping 38 CO<sub>2</sub> dynamics. Multivariate analysis identified soil organic carbon (SOC) and temperature as dominant flux drivers, and a potential SOC threshold was detected, 39 beyond which CO2 emissions accelerated. While intercropping reduced tea-row 40 emissions by 7.1-7.9% but increased inter-row emissions by 12.7-28.9%, continuous 41 intercropping significantly decreased overall inter-row emissions over time. These 42 results highlight the spatially heterogeneous nature of carbon flux regulation and 43 demonstrate the long-term potential of green manure intercropping as a climate-smart 44 45 management strategy in perennial agroecosystems.

**Keywords**: tea plantations, green manure, CO<sub>2</sub> emissions, soil factors

#### 1 Introduction

48

Mitigating greenhouse gas (GHG) emissions to address global warming and 49 associated climate challenges remains a priority in global environmental research. 50 51 Among long-lived GHGs, carbon dioxide (CO<sub>2</sub>) plays the most prominent role, contributing approximately 66% to the increase in global radiative forcing (IPCC, 52 2022). In 2023, the global average atmospheric CO<sub>2</sub> concentration reached  $420.0 \pm 0.1$ 53 ppm, representing a 151% increase compared to pre-industrial levels (prior to 1750) 54 (WMO, 2024). Agriculture is a major emission sector, accounting for about 14% of 55 total anthropogenic CO<sub>2</sub> emissions (Wang et al., 2025). In China, this share is even 56 higher, with agricultural activities accounting for up to 17% of national CO<sub>2</sub> emissions 57 (Xu and Lin, 2017). Therefore, accurately characterizing CO<sub>2</sub> emission dynamics in 58 agricultural systems and scientifically informed mitigation strategies are critical for 59 advancing global GHG reduction efforts and promoting sustainable, low-carbon 60 61 agricultural development (Xu et al., 2024). Tea (Camellia sinensis L.) is an important economic crop in tropical and 62 subtropical regions. Over recent decades, global tea cultivation area has expanded 63 64 rapidly, reaching 4.70 million hectares in 2022. China has led the most significant growth, with 3.35 million hectares of tea plantations and an annual production of 2.82 65 million tons, ranking first worldwide in both area and output (FAO, 2024). To maximize 66 67 yield and improve tea quality, fertilizer inputs to tea cultivation area can be up to four times higher than those applied to staple crops during a single growing season (Zou et 68 al., 2009; Han et al., 2013; Yao et al., 2015). In China, average annual fertilizer use in 69 70 tea plantations reaches 678 kg ha<sup>-1</sup>, with more than 30% of plantations experiencing over-application (Ni et al., 2019). Such intensive fertilization not only accelerates soil 71 acidification but also significantly increases GHGs emissions from tea plantations (Liu 72 et al., 2016; Yan et al., 2020). However, most existing studies on agricultural CO2 73 emissions have focused on staple cropping systems such as wheat (Song et al., 2024), 74 rice (Qian et al., 2023), and maize (Zhang et al., 2020), while studies on CO<sub>2</sub> emissions 75 from tea plantations remain limited. 76

For instance, Lang et al. (2017) reported that intercropping rubber trees with tea

simultaneously weakened the soil's methane (CH<sub>4</sub>) uptake capacity. Wanyama et al. 79 (2019) found that converting tropical montane forests into tea plantations in Africa 80 81 decreased annual soil CO<sub>2</sub> emissions to 5.6 t ha<sup>-1</sup>, with emissions positively correlated with soil pH and negatively correlated with the soil C/N ratio. Pang et al. (2019) 82 quantitatively assessed the net ecosystem exchange (NEE) of tea plantations in 83 southeastern China from 2014 to 2017, reporting values ranging from -182.40 to 84 -301.51 g C m<sup>-2</sup>, indicating that tea plantations act as net carbon sinks. However, their 85 carbon sequestration potential was lower than that of other subtropical ecosystems, with 86 temperature identified as the primary factor influencing ecosystem respiration. These 87 findings suggest that CO2 emissions from tea plantations play a non-negligible role on 88 the carbon exchange between atmosphere and tea plantations. However, the limited 89 number of studies has led to substantial uncertainty on estimating tea plantation CO2 90 91 emissions, restricting our understanding of their contribution to regional and global 92 agricultural GHG budgets (Li et al., 2016; Ji et al., 2020) and hindering the development 93 of low-carbon tea plantations. 94 In response, there is a growing emphasis on the development of eco-friendly and 95 low-carbon tea plantations (Wang et al., 2022). Toward reduction of fertilizer usage and 96 higher economic efficiency, various management practices were incorporated, 97 including by using green manure. As a modernized agricultural practice, green manure has been widely adopted in farming systems and serves as an important measure for 98 improving soil quality, playing a vital role in sustainable agriculture. Within the context 99 100 of GHGs mitigation, green manure is recognized as an effective solution for improving soil quality and enhancing CO<sub>2</sub> sequestration in agroecosystems (Forte et al., 2017). 101 However, most studies examining the relationship between green manure and carbon 102 emissions have focused on conventional croplands such as rice and wheat. 103 104 Comprehensive studies have shown that appropriate green manure management can significantly reduce the global warming potential (GWP) associated with fertilization 105 (Zhang et al., 2024). For instance, Gong et al. (2021) demonstrated that long-term 106 ryegrass cover in organic soybean fields effectively reduced net GWP. In contrast, other 107

in the tropical forests of Xishuangbanna, China, reduced CO2 emissions, although it

studies have reported that green manure application may increase CO<sub>2</sub> emissions. Kim et al. (2013) found that the application of Chinese milk vetch and ryegrass increased 109 winter CO<sub>2</sub> fluxes in paddy fields by approximately 197% and 266%, respectively. 110 Large-scale assessments have further revealed that green manure tends to increase CO<sub>2</sub> 111 emissions, primarily due to differences in plant species and biomass inputs. Biomass 112 alone explained 63% of the variation in CO2 emission increases, with emissions 113 declining as the C/N ratio of cover crop biomass increased. Notably, mixed sowing of 114 leguminous and non-leguminous green manures has been shown to improve residue 115 C/N ratios and reduce GHGs emissions (Muhammad et al., 2019). 116 Existing studies on GHG mitigation in tea systems have predominantly focused on 117 fertilizer reduction and substitution strategies. Wu et al. (2018) conducted a three-year 118 119 fertilization control experiment in southern China and found that halving nitrogen input decreased nitrous oxide (N2O) emissions by 44.5% in tea plantations. In contrast, Yao 120 121 et al. (2015) reported that organic fertilizer application led to a 71% increase in N<sub>2</sub>O emissions compared to conventional urea, suggesting potential trade-offs in GHG 122 123 outcomes. Organic amendments, such as compost or manure, have been shown to 124 improve soil fertility, enhance soil structure, porosity, and pH, and promote carbon 125 sequestration in tea plantation soils (Han et al., 2013; Wu et al., 2021). Biochar application has also been identified as an effective strategy for improving soil quality 126 127 while simultaneously enhancing soil carbon storage and reducing emissions (Wu et al., 2021). The effect of green manure intercropping on tea plantations was mainly focused 128 on improvements in tea plant growth and soil nutrient dynamics. For example, 129 130 intercropping with green manure species has been shown to enhance nitrogen use efficiency and increase soil microbial diversity (Huang et al., 2023). The potential role 131 of green manure intercropping in mitigating GHGs emissions in tea ecosystems remains 132 poorly understood, and its interactions with key environmental factors have not been 133 134 fully clarified (Zhu et al., 2018). Green manure may influence CO<sub>2</sub> emissions by altering carbon input levels and inducing soil disturbances, but the specific emission 135 characteristics and driving factors require further investigation. 136

To address these gaps, this study selected cultivated tea plantations region located

in the east of China, where is recognized as a very important tea cultivation area famous by the tea name of Longjing. Commonly used green manure species in tea systems (*Vulpia myuros* C., *Lolium perenne* L., and *Trifolium repens* L.) were selected for intercropping, covering both leguminous and non-leguminous species, under monoculture and mixed-sowing configurations. CO<sub>2</sub> flux were carried out in both tea rows and inter-row zones across different intercropping treatments. Dynamics and key influencing factors of CO<sub>2</sub> emissions under various green manure intercropping models were analyzed, aiming to reveal the potential reasons by which green manure intercropping regulates carbon fluxes in tea plantations and provide support for the development of low-carbon tea plantations and sustainable regional agriculture.

#### 2 Methodology

# 2.1 Monitoring Site

This study was conducted at the Comprehensive Experimental Tea Plantation Base of the Tea Research Institute, Chinese Academy of Agricultural Sciences, located in Shengzhou, Zhejiang Province, China (120°83′E, 29°75′N; elevation 30 m a.s.l.) (Fig. 1a). The site is situated in a low mountainous and hilly region of southeastern China and is characterized by a subtropical monsoon climate. During the experimental period (August 2022 to August 2024), the average annual temperature was approximately 16 °C, with an average annual precipitation of about 1400 mm. The region experiences a concentrated rainy season from April to June and has a frost-free period of around 240 days. The tea cultivated at the site is *Jinmudan*, an elite cultivar derived from the hybridization of *Tieguanyin* and *Huangdan*, and is widely planted across China. The soil type of the tea plantation is classified as red soil (Ultisol), typical soil species in this region.

## 2.2 Experimental Setup

Three green manure intercropping treatments were established in this study: *Vulpia myuros* C. (SM), a mixture of *Lolium perenne* L. and *Trifolium repens* L. (HM), and a control treatment without intercropping (CK). Each treatment has three duplications.

Gas fluxes were measured in both tea rows (T) and inter-row zones (G), resulting in six treatments: SMT, SMG, HMT, HMG, CKT, and CKG, with a total of 18 representative 169 sampling points (Fig. 1b, c). Tea plantation followed standard management practices, 170 including fertilization, pruning, and tillage. Basal fertilization and tillage were 171 conducted in middle of October, followed by green manure sowing in early November. 172 Urea was applied as a topdressing in early February. Tea leaves were harvested in end 173 of March, and pruning was normally conducted in May and July. 174 Gas sampling was performed using the static chamber-gas chromatography 175 method. The dimensions of the static chambers were  $1.25 \times 0.8 \times 1.0$  m for tea rows and 176  $0.3 \times 0.3 \times 0.5$  m for inter-row zones (Fig. 1b). Each chamber was equipped with an 177 internal fan to ensure uniform gas mixing. To avoid rapid heating due to sunlight, the 178 179 chambers were wrapped with aluminum foil and sponge, functioning as dark chambers. 180 To minimize disturbance, chamber bases with water grooves were installed one month 181 in advance at each sampling point, inserted 15 cm into the soil. During sampling, water 182 was added to the grooves, and the chamber was securely sealed onto the base to create a closed environment. Four gas samples were collected at 7-minute intervals using gas 183 184 sampling bags. For seasonal monitoring, sampling was conducted once per week between 9:00 185 and 11:00 a.m. (local time). Intensive sampling was also carried out following key 186 187 management events such as fertilization and pruning. Diurnal variation was monitored over three consecutive days in January, April, July, and October, representing winter, 188 189 spring, summer, and autumn, respectively. During these campaigns, gas samples were 190 collected every 2 hours over a 24-hour period. All gas samples were analyzed within 24 hours by using a gas chromatograph (Agilent 7890B, Agilent Inc., USA). CO<sub>2</sub> 191 concentrations were measured using a flame ionization detector (FID) at a working 192 temperature of 175 °C. High-purity nitrogen was used as the carrier gas, with an 193 injection volume of 30 mL and a flow rate of 250 mL·min<sup>-1</sup>. 194 Meteorological data, including precipitation, atmospheric pressure, and air 195 temperature (AT), were obtained from an automatic weather station installed within the 196 tea plantation. Soil samples were recorded using an automatic weather station installed 197

within the tea plantation. Soil samples were collected monthly using a five-point composite method within a 1 m radius of each sampling point. After passing through a 2 mm sieve, the samples were divided into three portions:

- One fresh portion was analyzed for microbial biomass carbon (MBC) and microbial biomass nitrogen (MBN) using the chloroform fumigation-extraction method and a TOC analyzer.
- ii. A second portion was air-dried and ground for analysis of soil pH.
  - iii. The third portion was stored at 4 °C for analysis of nitrate nitrogen (NO<sub>3</sub><sup>-</sup>-N) and ammonium nitrogen (NH<sub>4</sub><sup>+</sup>-N) by spectrophotometry, total carbon (TC) and total nitrogen (TN) by elemental analysis, and soil organic carbon (SOC) by the dichromate oxidation–spectrophotometry method.

Soil temperature (ST) and volumetric water content (VWC) were measured *in-situ* using a portable soil sensor (TDR-315H, Acclima). Soil bulk density (BD) and water-filled pore space (WFPS) were determined using the core ring method.

## 2.3 Data Processing

The flux refers to the amount of gas exchanged per unit time and unit area. A positive value indicates net emission to the atmosphere, while a negative value indicates net uptake from the atmosphere (Yao et al., 2015; Zhang et al., 2020). Based on the flux measurements, cumulative CO<sub>2</sub> emissions under different green manure intercropping treatments were also estimated. All data analyses and visualizations were performed using R software. Two-way analysis of variance (Two-way ANOVA) was employed to assess the effects of treatment type and observation period on CO<sub>2</sub> fluxes and soil physicochemical properties. Spearman correlation analysis and Mantel tests were used to examine the relationships between CO<sub>2</sub> flux and environmental variables under different green manure intercropping treatments. Canonical correspondence analysis (CCA) was applied to comprehensively evaluate the influence of soil physicochemical properties on CO<sub>2</sub> emissions.

228

# 3 Results

# 3.1 Long-term Variation of CO<sub>2</sub> Fluxes under Green Manure Intercropping

Figure 2 illustrates the long-term trends of key environmental variables and CO2 229 230 fluxes in the tea plantation throughout the observation period. Overall, CO<sub>2</sub> fluxes from both tea-row and inter-row zones displayed distinct seasonal patterns: higher in summer 231 232 and autumn, and lower in spring and winter. The seasonal differences between the warm (summer and autumn) and cool (spring and winter) periods were statistically significant 233 (Table 1). The temporal dynamics of CO<sub>2</sub> fluxes closely tracked the trends in air 234 temperature (Fig. 2a), suggesting that temperature is a key driver of soil respiration in 235 tea plantations. Annual fluctuations in CO2 fluxes were also strongly influenced by field 236 management activities. For example, a sharp increase in CO2 emissions was observed 237 238 following basal fertilization in October, and another rise occurred in March of the following year after topdressing and with the onset of warmer temperatures, ultimately 239 240 peaking in summer (Fig. 2b-c). The effects of management activities are further detailed 241 in Section 3.3. In tea rows, the annual mean CO2 fluxes under HMT and SMT treatments were 242 243  $7.35 \pm 0.44 \text{ mg} \cdot \text{m}^{-2} \cdot \text{min}^{-1}$  and  $7.41 \pm 0.45 \text{ mg} \cdot \text{m}^{-2} \cdot \text{min}^{-1}$ , respectively, both lower than 244 that of the control (CKT:  $8.12 \pm 0.46 \text{ mg} \cdot \text{m}^{-2} \cdot \text{min}^{-1}$ ) (Fig. 2b). In contrast, in inter-row 245 zones, the annual mean CO<sub>2</sub> fluxes were significantly higher under HMG (9.77 ± 0.54 246  $\text{mg} \cdot \text{m}^{-2} \cdot \text{min}^{-1}$ ) and SMG (10.83 ± 0.52  $\text{mg} \cdot \text{m}^{-2} \cdot \text{min}^{-1}$ ) compared to the control (CKG: 9.07 ± 0.44 mg·m<sup>-2</sup>·min<sup>-1</sup>) (Fig. 2c). Across seasons, CKT generally exhibited higher 247 CO2 fluxes than HMT and SMT, except during winter. In the inter-row zones, both 248 249 HMG and SMG showed significantly higher fluxes than CKG in summer, while SMG consistently had significantly higher CO<sub>2</sub> emissions than both CKG and HMG during 250 the remaining seasons (p < 0.05) (Table 1). 251 Overall, green manure intercropping significantly increased CO2 emissions from 252 253 inter-rows, but reduced emissions in tea rows. In terms of cumulative annual emissions, HMT and SMT resulted in 3.69 kg·m<sup>-2</sup> and 3.66 kg·m<sup>-2</sup> of CO<sub>2</sub> emissions, respectively, 254 both lower than the 3.97 kg·m<sup>-2</sup> under CKT (Fig. 3). Similarly, cumulative CO<sub>2</sub> 255 emissions under HMG and SMG remained consistently higher than under CKG, but 256

they declined from 5.76 kg·m<sup>-2</sup> and 6.43 kg·m<sup>-2</sup> in the first year to 4.16 kg·m<sup>-2</sup> and 4.92 kg·m<sup>-2</sup> in the second year, respectively (Fig. 3). Two consecutive years of green manure intercropping led to a gradual reduction in CO<sub>2</sub> emissions from inter-rows, indicating its potential role in long-term emission mitigation in tea plantations. CO<sub>2</sub> emissions from inter-rows were substantially higher than those from tea rows. Compared with the control, HM and SM intercropping increased inter-row cumulative CO<sub>2</sub> emissions by 12.7% and 28.9%, respectively, while reducing tea-row emissions by 7.1% and 7.9% (Fig. 3a-b). Inter-row zones accounted for 52.6%, 57.3%, and 60.8% of the total annual CO<sub>2</sub> emissions in the CK, HM, and SM treatments, respectively (Fig. 3c-d), indicating that the inter-row emissions cannot be ignored.

266267

268

269270

271272

278279

282283

259260

261262

#### 3.2 Diurnal CO<sub>2</sub> Variations

CO2 fluxes in the tea plantation exhibited pronounced diurnal variations across all seasons, particularly in spring and summer (Fig. 4), likely influenced by the growth stages of green manure species. In spring, CO2 fluxes in tea rows under all treatments showed a similar diurnal trend: an initial decline followed by a rapid increase. HMT and SMT reached their minimum fluxes at 08:00 (local time), with values of -3.74 mg·m<sup>-2</sup>·min<sup>-1</sup> and -3.80 mg·m<sup>-2</sup>·min<sup>-1</sup>, respectively, then rose sharply and stabilized in the afternoon. The diurnal amplitudes under HMT and SMT were notably greater than that of the control (CKT) (Fig. 4a). In the inter-row zones, the diurnal patterns under green manure treatments differed notably from the control (Fig. 4b). CKG displayed a unimodal pattern with a peak at 12:00 (12.74 mg·m<sup>-2</sup>·min<sup>-1</sup>) and a trough at 08:00 (5.45 mg·m<sup>-2</sup>·min<sup>-1</sup>), resulting in an amplitude of 7.29 mg·m<sup>-2</sup>·min<sup>-1</sup>. In contrast, HMG and SMG exhibited later peaks at 16:00 (23.26 mg·m<sup>-2</sup>·min<sup>-1</sup>) and 14:00 (24.17 mg·m<sup>-2</sup>·min<sup>-1</sup>), respectively, with troughs also at 08:00 (HMG: 12.28 mg·m<sup>-2</sup>·min<sup>-1</sup>; SMG: 12.43 mg·m<sup>-2</sup>·min<sup>-1</sup>). Both treatments showed substantially higher amplitudes than CKG. Summer exhibited the most pronounced diurnal variation of CO<sub>2</sub> fluxes across all seasons. In tea rows, CKT, HMT, and SMT followed a bimodal pattern, with peaks at

287 6.70, and 10.10 mg·m<sup>-2</sup>·min<sup>-1</sup> (Fig. 4c). In the inter-rows, the amplitudes were relatively lower, 7.72, 8.12, and 7.79 mg·m<sup>-2</sup>·min<sup>-1</sup> for CKG, HMG, and SMG, 288 respectively, indicating smaller fluctuations compared to tea rows (Fig. 4d). Notably, 289 290 summer also showed the most distinct contrast between tea rows and inter-rows: CKT recorded the highest average flux in the tea rows, while CKG had the lowest in the inter-291 292 rows. In autumn, tea-row fluxes under all treatments exhibited a unimodal pattern, with 293 minima at 08:00 and peaks at 14:00. The diurnal amplitudes were 11.27, 8.02, and 12.75 294 mg·m<sup>-2</sup>·min<sup>-1</sup> for CKT, HMT, and SMT, respectively (Fig. 4e). In the inter-rows, HMG 295 and SMG displayed relatively stable diurnal trends, whereas CKG showed a bimodal 296 pattern with peaks at 06:00 and 16:00, and a greater amplitude than both HMG and 297 298 SMG (Fig. 4f). In winter, CO2 fluxes showed the most stable diurnal variation of the year. In tea 299 300 rows, amplitudes were only 2.96, 2.84, and 4.92 mg·m<sup>-2</sup>·min<sup>-1</sup> for CKT, HMT, and SMT, respectively (Fig. 4g). Unlike other seasons, 08:00 no longer corresponded to the 301 302 daily minimum but rather to a relative maximum, with daily peaks generally occurring 303 at 14:00. In inter-rows, diurnal patterns were less defined. SMG exhibited the highest 304 flux at 08:00 (7.57 mg·m<sup>-2</sup>·min<sup>-1</sup>), while HMG showed the lowest at 14:00 (0.99

314

# 3.3 Effect of Human Management on CO<sub>2</sub> Fluxes

mg·m<sup>-2</sup>·min<sup>-1</sup>) (Fig. 4h).

CO<sub>2</sub> fluxes from the tea field varied significantly across different growth stages of green manure, exhibiting a general increasing trend from the early growth stage to the vigorous, wilting, and decomposition stages (Fig. 5a). In the tea rows, the lowest fluxes were observed during the early growth stage, while the highest occurred during the decomposition stage. Differences among the three treatments (CKT, HMT, and SMT) were minimal during the early growth but became more apparent in the subsequent stages. Notably, during the vigorous stage, both HMT and SMT treatments reduced CO<sub>2</sub> emissions compared to CKT. In contrast, the impact of green manure growth on CO<sub>2</sub> fluxes was more pronounced in the inter-row zones (Fig. 5b). At all growth stages, CO<sub>2</sub>

fluxes under the HMG and SMG treatments were significantly higher than those under CKG, with the largest differences observed during the wilting stage. Peak emissions 318 occurred during the decomposition stage for HMG and during the wilting stage for 319 320 SMG. Fertilization substantially increased CO<sub>2</sub> emissions across the tea plantation (Fig. 321 5c-d). In tea rows, the post-fertilization increase in CO<sub>2</sub> flux under HMT was 43.1% 322 323 lower than that under CKT, whereas SMT showed a 9.2% higher increase. In the interrow zones, HMG reduced the fertilization-induced increase by 10.4% compared to 324 CKG, while SMG amplified it by 40.1%. These findings indicate that the HM treatment 325 can effectively mitigated CO2 emissions triggered by fertilization, while SM treatment 326 may intensify them, revealing the potential of legume-based mixed green manure to 327 reduce CO<sub>2</sub> emissions in tea plantations. It is worth noting that the mitigation effect in 328 the inter-row zones was weaker than in the tea rows, possibly due to differences in root 329 330 distribution or organic matter inputs. The effects of grass planting and tea pruning on CO2 fluxes varied by treatment 331 332 type and location (tea row or inter-row) (Fig. 5c-d). In the CK treatment, grass planting 333 had no significant impact on CO2 fluxes. However, the HM treatment led to a marked 334 increase after grass planting, with inter-row fluxes rising by 1.81 mg·m<sup>-2</sup>·min<sup>-1</sup>. 335 Similarly, the SM treatment showed significant increases in both zones, with an 336 increase of 0.90 mg·m<sup>-2</sup>·min<sup>-1</sup> in tea rows and an inter-row increase that was 3.8 times greater. These increases can be attribute to soil disturbance during sowing. 337 After tea pruning, no significant changes in CO2 flux were observed in the CK 338 339 treatment. However, both HMT and SMT significantly increased CO<sub>2</sub> emissions in tea row, with increments of 2.74 mg·m<sup>-2</sup>·min<sup>-1</sup> and 2.94 mg·m<sup>-2</sup>·min<sup>-1</sup>, respectively. In the 340 inter-row zones, only the HMG treatment exhibited a significant post-pruning increase 341 of 3.25 mg·m<sup>-2</sup>·min<sup>-1</sup>. These increases may be attributed to pruning residues covering 342 343 the green manure surface, which could elevate soil temperature and moisture, thereby 344 enhancing soil respiration and CO<sub>2</sub> emissions.

351

357

359

362

368

371372

#### 3.4 Effects of Environmental Factors on CO<sub>2</sub> Fluxes

Significant differences in soil nutrient parameters were observed between tea rows and inter-row zones under various green manure intercropping treatments (Fig. 6). Green manure treatments generally increased soil total carbon (TC) and total nitrogen (TN), with consistently higher TC and TN levels in the inter-row zones than in the tea rows (Fig. 6a-b), resulting in significantly higher C/N ratios in the tea rows (Fig. 6c). Soil ammonium nitrogen (NH<sub>4</sub><sup>+</sup>-N) and nitrate nitrogen (NO<sub>3</sub><sup>-</sup>-N) concentrations were also significantly greater in the inter-row zones, with the highest NH<sub>4</sub><sup>+</sup>-N found in CKG (71.20 mg·kg<sup>-1</sup>) and the highest NO<sub>3</sub><sup>-</sup>-N in SMG (14.56 mg·kg<sup>-1</sup>). All green manure treatments significantly increased soil organic carbon (SOC), the average SOC contents under HM and SM were 3.6% and 9.3% higher than under CK, respectively (Fig. 6f). Microbial biomass carbon (MBC) and microbial biomass nitrogen (MBN) showed no significant differences between tea rows and inter-row zones, but both were slightly elevated under green manure treatments (Fig. 6g-h). Soil pH ranged from 3.6 to 4.5, with no significant differences among treatments, although green manure application slightly increased soil pH (Fig. 6i). During the monitoring, soil temperature ranged from 2.3–41.8°C in CK, 3.0– 37.6°C in HM, and 3.5–36.6°C in SM. Average soil temperatures in HM and SM were 4.5% and 3.9% lower than in CK, indicating a cooling effect of green manure. Additionally, bulk density was reduced by 8.9% and 5.0% in HM and SM compared to CK, while total porosity increased by 5.3% and 3.0%, and WFPS decreased by 29.1% and 11.1%, respectively. These results suggest that green manure intercropping effectively reduces soil compaction and improves soil aeration. The combined effect of these factors is the key to the changes in CO<sub>2</sub> emissions. To further clarify these relationships, we examined the correlations between CO<sub>2</sub> fluxes and environmental factors under different green manure treatments (Fig. 7). CO2 fluxes across nearly all treatments were significantly positively correlated with air temperature (AT) and soil temperature (ST) (r > 0.5, p 

rows and inter-row zones showed negative correlations with CO2 flux, whereas volumetric water content (VWC) showed significant positive correlation (r > 0.5, p <377 0.05). In contrast, VWC was negatively correlated with CO<sub>2</sub> flux under CK treatment. 378 379 These findings suggest that green manure intercropping alters soil pore structure and moisture regimes, thereby modifying CO<sub>2</sub> emission dynamics compared to bare soil 380 conditions. Additionally, environmental controls on CO2 fluxes differed between tea 381 rows and inter-row zones. Emissions in tea rows appeared less sensitive to 382 environmental fluctuations, likely due to the moderating effects of tea canopy coverage 383 and root systems. 384 Canonical correspondence analysis (CCA) was further performed to examine the 385 effect of green manure intercropping patterns and soil properties on CO2 emissions in 386 tea rows and inter-row zones (Fig. 8). The first two CCA axes explained 52.79% and 387 11.15% of the total variance, respectively. CCA1 was primarily associated with NH<sub>4</sub>+-388 389 N, NO<sub>3</sub>-N, SOC, and the C/N ratio, while CCA2 was mainly linked to TN, TC, MBN, MBC, and pH. These results indicate distinct environmental drivers of CO<sub>2</sub> emissions 390 391 between the two spatial zones. In tea rows, CO2 flux was positively associated with 392 NO<sub>3</sub>-N and the C/N ratio, with relatively minor influence from pH. The influencing 393 soil factors were similar for CKT and SMT, whereas HMT displayed a distinct pattern, 394 likely attributable to the presence of leguminous green manure. In inter-row zones, SOC 395 emerged as the dominant factor controlling CO<sub>2</sub> fluxes in HMG and SMG treatments, whereas NH<sub>4</sub>+-N was the key driver in CKG. Moreover, TC, TN, MBC, and MBN all 396 showed positive associations with inter-row CO2 fluxes, with consistent soil drivers 397 398 under HMG and SMG that differed from CKG, indicating that green manure significantly affects soil-CO2 interactions. 399

403

### 4 Discussion

# 4.1 CO<sub>2</sub> Flux Dynamics under Green Manure Intercropping

This study revealed pronounced seasonal variations in CO2 fluxes from tea plantations, which were closely aligned with fluctuations in air temperature (Fig. 2a-c). In spring, rising temperatures enhanced both plant and microbial respiration, leading to

406 a sharp increase in CO<sub>2</sub> emissions (Yan et al., 2022). During summer, when temperatures reached their annual peak, intensified microbial activity accelerated the 407 decomposition of soil organic matter, resulting in the highest CO2 fluxes of the year 408 409 (Allison et al., 2010). In autumn, declining temperatures and light availability reduced microbial activity and soil respiration, thereby lowering CO<sub>2</sub> emissions (Liu et al., 410 2020). In winter, low temperatures significantly inhibited both plant and microbial 411 respiration, causing CO2 fluxes to drop to their annual minimum (Schnecker et al., 412 2023). These seasonal flux patterns were consistent throughout the two-year 413 observation period, indicating the pivotal role of temperature in regulating CO<sub>2</sub> 414 emissions in tea plantations (Chen et al., 2021). 415 Compared with the CK treatment, HM and SM increased annual cumulative CO<sub>2</sub> 416 emissions by 3.3% and 7.9%, respectively, revealing that green manure intercropping 417 significantly elevated total CO2 emissions. Similar findings have been reported in 418 419 previous studies. For example, Lee et al. (2021) observed consistently higher 420 cumulative CO<sub>2</sub> emissions in cropland soils under green manure treatments than under 421 fallow conditions. A meta-analysis by Muhammad et al. (2019) also showed that the 422 use of cover crops generally increases CO<sub>2</sub> emissions compared with bare soil. This 423 effect can be attributed to two possible mechanisms: 1) green manure crops introduce exogenous carbon inputs into the soil, which stimulates CO<sub>2</sub> release (Ho et al., 2021); 424 425 and 2) green manure intercropping reduces soil bulk density and increases total porosity (Song et al., 2016), thereby improving soil aeration and promoting aerobic microbial 426 activity. The enhanced microbial activity accelerates the decomposition and 427 428 mineralization of soil organic matter, consequently increasing CO<sub>2</sub> emissions (Chen et al., 2019). By contrast, under the CK treatment, higher WFPS and greater soil 429 compaction may have inhibited gas diffusion and limited CO2 release into the 430 atmosphere (Lang et al., 2017). 431 Diurnal variations of CO<sub>2</sub> fluxes were influenced by both seasonal dynamics and 432 the growth stages of green manure crops. CO<sub>2</sub> fluxes fluctuated most sharply during 433 spring and summer, with temperature identified as the primary driver of daily flux 434 patterns (Pang et al., 2019). In spring, negative CO2 flux peaks were observed in tea 435

446447

448449

rows under HMT and SMT around 08:00, due to low morning temperatures suppressing microbial respiration. Moreover, the presence of easily degradable organic matter from green manure may have diverted microbial metabolism toward biomass accumulation rather than complete mineralization to CO2. In summer, distinct spatial differences appeared between tea rows and inter-row zones. In tea rows, CO<sub>2</sub> fluxes under the CKT treatment were significantly higher than those under HMT and SMT, while in the interrow zones, fluxes under HMG and SMG were higher than those under CKG. This spatial heterogeneity highlights the dual role of green manure: in tea rows, the shading effect of green manure canopy reduced soil temperatures, thereby inhibiting microbial respiration; in contrast, inter-row zones were exposed to direct sunlight, root exudates and decomposing plant residues provided additional carbon sources. Under favorable thermal conditions, this stimulated microbial activity and thus increased CO<sub>2</sub> emissions (Gui et al., 2024). In autumn and winter, CO<sub>2</sub> flux peaks were mostly recorded in the afternoon, possibly due to rising temperatures reaching a threshold that accelerated enzymatic reactions and microbial metabolism, enhancing root and soil respiration and thus elevating CO<sub>2</sub> emissions (Dove et al., 2021).

451452453

457

#### 4.2 Influence of Cultivation Management on CO2 Fluxes

Fertilization, pruning, and soil tillage with cover cropping are critical anthropogenic management practices in tea plantations that significantly affect CO<sub>2</sub> flux dynamics. Fertilizer application, in particular, is a major contributor to agricultural greenhouse gas emissions, with emission strength influenced by the type, amount, and method of application (Wang et al., 2024). In this study, the application of rapeseed cake and compound fertilizers significantly increased soil CO<sub>2</sub> fluxes, especially within tea rows. This increase can be attributed to two main factors: 1) the input of exogenous organic matter enriched soil organic carbon content; and 2) trench fertilization caused physical disturbance, disrupting soil aggregates and accelerating the decomposition of soil organic carbon. These disturbances stimulated the abundance and metabolic activity of aerobic heterotrophic microbes, promoting organic matter mineralization and resulting in elevated CO<sub>2</sub> emissions (Chappell et al., 2015; Struck et al., 2020).

477

479

482

485

487

493

fluxes. Pruning substantially reduces the photosynthetic biomass of tea plants, diminishing their carbon sequestration capacity. Simultaneously, the resulting litterfall provides abundant substrates for microbial respiration (Pang et al., 2019). The combined effect of reduced carbon uptake and increased decomposition substrates leads to a rapid short-term increase in CO<sub>2</sub> emissions after pruning. Previous studies have demonstrated that intercropping systems introduce readily decomposable carbon through root exudates, while green manure decomposition increases organic matter inputs and improves soil organic carbon storage (Gui et al., 2024). In this study, CO<sub>2</sub> fluxes in inter-row zones were significantly higher than the control during the wilting and decomposition stages of green manure, suggesting that microbial activity was enhanced during these periods, thereby accelerating the decomposition and transformation of organic matter and intensifying soil respiration. Additionally, soils under green manure treatments exhibited lower annual average temperatures compared to the control, indicating that intercropping with green manure moderated surface soil temperatures and reduced daily temperature fluctuations. This effect was particularly pronounced in summer, when green manure not only reduced inter-row CO<sub>2</sub> emissions but also improved the microclimatic conditions of the tea plantation. Traditional CO<sub>2</sub> flux measurements in tea plantations have mostly focused only on tea rows, often neglecting inter-row soil emissions (Yao et al., 2015; Chen et al., 2021). In this study, static chambers were parallelly employed to measure CO2 fluxes in tea rows and inter-row areas, enabling a more accurate understanding of CO<sub>2</sub> emissions. Results showed that inter-row CO<sub>2</sub> fluxes were significantly higher than those in tea rows (p < 0.05), accounting for 52.6%, 57.3%, and 60.8% of the annual cumulative CO<sub>2</sub> emissions under CK, HM, and SM treatments, respectively. These findings emphasize the substantial contribution of inter-row zones to overall CO<sub>2</sub> emissions. This discrepancy is due to differences in management intensity: fertilization and tillage are commonly performed in inter-row areas, while the soil beneath tea canopies experiences minimal disturbance (Hirono and Nonaka, 2012). Moreover, pruning

Intensive pruning conducted in May and August further contributed to increased CO2

residues often accumulate in inter-row zones, further intensifying microbial activity and CO<sub>2</sub> emissions in these areas. The cumulative CO<sub>2</sub> emissions under HMG and SMG treatments were significantly lower in the second year. This reduction may be attributed to reduced human disturbance and the regulatory effects of green manure (Gui et al., 2024). Therefore, long-term and systematic monitoring of inter-row soil CO<sub>2</sub> emissions is essential for accurately assessing the carbon dynamics and mitigation potential of tea plantation ecosystems.

502503

508509

511512

514515

516517

521522

496

497

498 499

500501

### 4.3 Differences in Environmental Drivers

Soil CO2 fluxes are regulated by multiple environmental factors, including photosynthetic activity or vegetation productivity (Tang et al., 2005), and soil properties such as temperature and moisture (Liu et al., 2023; Widanagamage et al., 2025). Among them, temperature is widely recognized as a primary driver of seasonal variation in soil respiration (Lang et al., 2017). Our results showed that CO<sub>2</sub> fluxes in both tea rows and inter-row areas were significantly correlated with soil and air temperatures under different green manure treatments (Fig. 7). In addition, carbon and nitrogen transformation processes driven by microorganisms are closely coupled. Nitrification and denitrification alter NO<sub>3</sub>-N and NH<sub>4</sub>+N levels, thereby influencing soil physicochemical properties and microbial activity. As a result, CO2 emissions exhibit significant positive correlations with nitrogen mineralization, denitrification, and N<sub>2</sub>O emissions (Dai et al., 2020). This carbon-nitrogen coupling may interact with the distinctive nutrient uptake characteristics of tea plants, which are ammonium-preferring species with rapid NH<sub>4</sub><sup>+</sup> assimilation (Xin et al., 2024). In our study, CO<sub>2</sub> fluxes under the CKG treatment were positively correlated with NH<sub>4</sub>+-N content (Fig. 9). NH<sub>4</sub>+-N levels under this treatment (71.20 mg·kg<sup>-1</sup>) were significantly higher than in the SMG and HMG treatments, whereas the corresponding soil pH value (3.97) was significantly lower (p < 0.05). This concurrent high NH<sub>4</sub><sup>+</sup>-N level and strong acidification is due to ammonium accumulation under conventional fertilization and subsequent H<sup>+</sup> release during nitrification (Chen et al., 2021). By contrast, the soil pH under green manure intercropping treatments increased by 0.02-0.11 units compared to the CK treatment

buffered soil acidification by reducing H<sup>+</sup> release during NH<sub>4</sub><sup>+</sup> nitrification. Moreover, 527 the NO<sub>3</sub>-N concentration in the SMG treatment (14.56 mg·kg<sup>-1</sup>) was significantly 528 529 higher than that in other treatments (Fig. 6e), due to the high biomass of Vulpia myuros C., which may reduce nitrate losses via runoff or leaching. Its active root system also 530 improved soil aeration, inhibiting denitrification under anoxic conditions. 531 SOC content reflects the dynamic balance between organic matter inputs and 532 decomposition (Mo et al., 2024). In our study, SOC levels in the HMT and SMT 533 treatments were higher than those in the CKT treatment, while their cumulative annual 534 CO2 emissions were lower. This indicates that increasing SOC storage can help mitigate 535 greenhouse gas emissions, consistent with findings by Han et al. (2022). However, the 536 HMG and SMG treatments exhibited much higher SOC levels than CKG, while their 537 cumulative CO2 emissions exceeded those of CKG. This implies that once SOC 538 539 accumulation surpasses a certain threshold, the excess carbon supply may stimulate microbial activity and subsequently increase CO<sub>2</sub> emissions (Lim and Choi, 2014). 540 Interestingly, recent studies reveal that SOC thresholds can modulate the impact of 541 542 nitrogen fertilization on carbon sequestration. In SOC-poor soils, nitrogen inputs tend to promote carbon accumulation and soil aggregation, enhancing SOC storage. 543 Conversely, in SOC-rich soils, nitrogen fertilization may enhance microbial metabolic 544 545 efficiency and increase microbial residue production (Ling et al., 2025). These insights provide a new perspective for interpreting our results and highlight the importance of 546 identifying threshold values under multifactorial interactions to better assess their 547 548 effects on CO2 emissions. In tea rows, excessively high soil C/N ratios may result in nitrogen limitation, 549 thereby inhibiting rapid decomposition of organic matter and reducing CO2 fluxes. 550 Green manure, as a fresh plant residue with a relatively low C/N ratio, can be rapidly 551 decomposed by soil microbes after incorporation, thus maintaining or enhancing SOC 552 553 levels (Li et al., 2024), which aligns with our observations (Fig. 7d). MBC and MBN are generally considered closely linked to SOC (Gao et al., 2022). However, in our 554 study, a significant positive correlation between MBC and SOC was only observed in 555

(Fig. 6i), suggesting that root exudates and organic matter inputs from green manure

the CKT treatment. The lack of correlation under green manure treatments may be due to the rapid and excessive input of exogenous carbon, which complicates the relationship between these variables. No significant differences in MBC and MBN levels were found among green manure treatments, and both showed weak correlations with CO<sub>2</sub> fluxes (Fig. 8), suggesting that MBC and MBN are not key drivers of CO<sub>2</sub> emissions in tea plantations.

#### **5 Conclusion**

This study revealed the regulation of CO<sub>2</sub> fluxes in tea plantations under different green manure intercropping treatments. Green manure significantly influenced CO<sub>2</sub> flux dynamics, with pronounced seasonal variations, higher fluxes in summer and autumn and lower fluxes in spring and winter. CO<sub>2</sub> emissions from inter-row areas were consistently higher than those from tea rows. CO<sub>2</sub> fluxes in the SM and HM treatments were significantly lower than in the CK treatment within tea rows, while the opposite trend was observed in inter-row areas, suggesting distinct spatial responses to green manure intercropping. Over the observation period, the HM and SM treatments reduced CO<sub>2</sub> emissions from tea rows by 7.1%–7.9%, while increasing inter-row emissions by 12.7%–28.9%. Inter-row CO<sub>2</sub> emissions accounted for 52.6%, 57.3%, and 60.8% of the annual cumulative CO<sub>2</sub> fluxes under the CK, HM, and SM treatments, respectively. These findings highlight the importance of incorporating spatial emission weighting into carbon accounting for agricultural ecosystems.

The observed spatial differences in CO<sub>2</sub> fluxes were closely related to variations in SOC content. Our findings suggest the existence of a critical SOC threshold that determines whether CO<sub>2</sub> emissions increase or decrease. Future research should focus on quantifying such thresholds under multi-factor interactions to better assess their impacts on greenhouse gas dynamics. Continuous green manure intercropping over two years significantly mitigated CO<sub>2</sub> emissions from inter-row areas. The HM treatment exhibited distinct diurnal CO<sub>2</sub> flux patterns and effectively suppressed fertilization-induced CO<sub>2</sub> emissions. These results demonstrate that green manure intercropping, particularly mixed legume and non-legume combinations, not only modifies the spatial

pattern of CO<sub>2</sub> emissions in tea plantations but also provides a practical strategy for mitigating carbon losses from managed agroecosystems. 587 588 589 590 Data availability The datasets generated and analyzed during this study are available from the 591 592 corresponding author upon reasonable request. 593 594 **Author contributions** 595 S. Liu conceived and designed the study, performed the data analysis, and drafted the 596 manuscript. S. Liu, Z. Jin, and Z. Chen contributed to data visualization. H. Li, Z. Fan, 597 S. Li, H. Fu, K. Zang, W. He, and P. Yan conducted field and laboratory work. S. Fang 598 supervised the research, provided funding, and contributed to manuscript review and 599 600 editing. 601 602 603 **Competing interests** 604 The contact author has declared that none of the authors has any competing interests. 605 606 Acknowledgments 607 608 This study was funded by the National Natural Science Foundation of China (42307126), the National Key Research and Development Program of China 609 (2023YFC3705205), Key Scientific Research Project of Tea Research Institute, 610 Chinese Academy of Agricultural Sciences (TRI-ZDRW-01-04), Open Fund Project of 611 Key Open Laboratory of Ecosystem Carbon Sources and Sinks (ECSS-CMA202309), 612 and the Zhejiang Provincial Research Development Program (2024C03246). We also 613 thanks to the staff who have contributed to the flux measurements at Shengzhou. 614

| 615 | References                                                                                                                                |  |  |  |  |  |
|-----|-------------------------------------------------------------------------------------------------------------------------------------------|--|--|--|--|--|
| 616 | Allison, S.D., Wallenstein, M.D., and Bradford, M.A.: Soil-carbon response to warming                                                     |  |  |  |  |  |
| 617 | dependent on microbial physiology, Nat. Geosci., 3, 336-340,                                                                              |  |  |  |  |  |
| 618 | http://doi.org/10.1038/ngeo846, 2010.                                                                                                     |  |  |  |  |  |
| 619 | Chappell, A., Baldock, J., and Sanderman, J.: The global significance of omitting soil                                                    |  |  |  |  |  |
| 620 | erosion from soil organic carbon cycling schemes, Nat. Clim. Chang., 6, 187-                                                              |  |  |  |  |  |
| 621 | 191, <a href="http://doi.org/10.1038/nclimate2829">http://doi.org/10.1038/nclimate2829</a> , 2015.                                        |  |  |  |  |  |
| 622 | Chen, D., Li, Y., Wang, C., Liu, X., Wang, Y., Shen, J., Qin, J., and Wu, J.: Dynamics                                                    |  |  |  |  |  |
| 623 | and underlying mechanisms of N2O and NO emissions in response to a transient                                                              |  |  |  |  |  |
| 624 | land-use conversion of Masson pine forest to tea field, Sci. Total Environ., 693,                                                         |  |  |  |  |  |
| 625 | 133549, http://doi.org/10.1016/j.scitotenv.2019.07.355, 2019.                                                                             |  |  |  |  |  |
| 626 | Chen, D., Wang, C., Li, Y., Liu, X., Qin, J., and Wu, J.: Effects of land-use conversion                                                  |  |  |  |  |  |
| 627 | from Masson pine forests to tea plantations on net ecosystem carbon and                                                                   |  |  |  |  |  |
| 628 | greenhouse gas budgets. Agric. Ecosyst. Environ., 320, 107578,                                                                            |  |  |  |  |  |
| 629 | http://doi.org/10.1016/j.agee.2021.107578, 2021.                                                                                          |  |  |  |  |  |
| 630 | Dai, Z., Yu, M., Chen, H., Zhao, H., Huang, Y., Su, W., Xia, F., Chang, S., Brookes, P.,                                                  |  |  |  |  |  |
| 631 | Dahlgren, R., and Xu, J.: Elevated temperature shifts soil N cycling from                                                                 |  |  |  |  |  |
| 632 | microbial immobilization to enhanced mineralization, nitrification and                                                                    |  |  |  |  |  |
| 633 | denitrification across global terrestrial ecosystems, Glob. Change Biol., 26,                                                             |  |  |  |  |  |
| 634 | 5267–5276, http://doi.org/10.1111/gcb.15211, 2020.                                                                                        |  |  |  |  |  |
| 635 | Dove, N., Torn, M.S., Hart, S.C., and Tas, N.: Metabolic capabilities mute positive                                                       |  |  |  |  |  |
| 636 | response to direct and indirect impacts of warming throughout the soil profile,                                                           |  |  |  |  |  |
| 637 | Nat. Commun., 12, 2089, <a href="http://doi.org/10.1038/s41467-021-22408-5">http://doi.org/10.1038/s41467-021-22408-5</a> , 2021.         |  |  |  |  |  |
| 638 | FAO, Statistical databases: <a href="https://www.fao.org/faostat/en/#data/">https://www.fao.org/faostat/en/#data/</a> , last accessed: 11 |  |  |  |  |  |
| 639 | June 2025.                                                                                                                                |  |  |  |  |  |
| 640 | Gao, D., Bai, E., Wang, S., Zong, S., Liu, Z., Fan, X., Zhao, C., and Hagedorn, F.:                                                       |  |  |  |  |  |
| 641 | Three-dimensional mapping of carbon, nitrogen, and phosphorus in soil                                                                     |  |  |  |  |  |
| 642 | microbial biomass and their stoichiometry at the global scale, Glob. Change                                                               |  |  |  |  |  |
| 643 | Biol., 28, 6728–6740, http://doi.org/10.1111/gcb.16374, 2022.                                                                             |  |  |  |  |  |
| 644 | Gong, Y., Li, P., Sakagami, N., and Komatsuzaki, M.: No-tillage with rye cover crop                                                       |  |  |  |  |  |

| 645 | can reduce net global warming potential and yield-scaled global warming                         |  |  |  |  |  |
|-----|-------------------------------------------------------------------------------------------------|--|--|--|--|--|
| 646 | potential in the long-term organic soybean field, Soil Tillage Res., 205, 104747,               |  |  |  |  |  |
| 647 | http://doi.org/10.1016/j.still.2020.104747, 2021.                                               |  |  |  |  |  |
| 648 | Gui, D., Zhang, Y., Lv, J., Guo, J., and Sha, Z.: Effects of intercropping on soil              |  |  |  |  |  |
| 649 | greenhouse gas emissions - A global meta-analysis, Sci. Total Environ., 918,                    |  |  |  |  |  |
| 650 | 170632, http://doi.org/10.1016/j.scitotenv.2024.170632, 2024.                                   |  |  |  |  |  |
| 651 | Han, W., Xu, J., Wei, K., Shi, Y., and Ma, L.: Estimation of N <sub>2</sub> O emission from tea |  |  |  |  |  |
| 652 | garden soils, their adjacent vegetable garden and forest soils in eastern China,                |  |  |  |  |  |
| 653 | Environ. Earth Sci., 70, 2495–2500, http://doi.org/10.1007/s12665-013-2292-4,                   |  |  |  |  |  |
| 654 | 2013.                                                                                           |  |  |  |  |  |
| 655 | Han, Z., Lin, H., Xu, P., Li, Z., Wang, J., and Zou, J.: Impact of organic fertilizer           |  |  |  |  |  |
| 656 | substitution and biochar amendment on net greenhouse gas budget in a tea                        |  |  |  |  |  |
| 657 | plantation, Agric. Ecosyst. Environ., 326, 107779,                                              |  |  |  |  |  |
| 658 | http://doi.org/10.1016/j.agee.2021.107779, 2022.                                                |  |  |  |  |  |
| 659 | Hirono, Y., and Nonaka, K.: Nitrous oxide emissions from green tea fields in Japan:             |  |  |  |  |  |
| 660 | contribution of emissions from soil between rows and soil under the canopy of                   |  |  |  |  |  |
| 661 | tea plants, Soil Sci. Plant Nutr., 58, 384–392,                                                 |  |  |  |  |  |
| 662 | http://doi.org/10.1080/00380768.2012.686434, 2012.                                              |  |  |  |  |  |
| 663 | Huang, X., Zheng, Y., Li, P., Cui, J., Sui, P., Chen, Y., and Gao, W.: Organic                  |  |  |  |  |  |
| 664 | management increases beneficial microorganisms and promotes the stability of                    |  |  |  |  |  |
| 665 | microecological networks in tea plantation soil, Front. Microbiol., 14, 1237842,                |  |  |  |  |  |
| 666 | http://doi.org/10.3389/fmicb.2023.1237842, 2023.                                                |  |  |  |  |  |
| 667 | IPCC: Climate Change 2022: Impacts, Adaptation, and Vulnerability, contribution of              |  |  |  |  |  |
| 668 | Working Group II to the Sixth Assessment Report of the Intergovernmental                        |  |  |  |  |  |
| 669 | Panel on Climate Change, Cambridge University Press, Cambridge, 2022.                           |  |  |  |  |  |
| 670 | Ji, C., Li, S., Geng, Y., Yuan, Y., Zhi, J., Yu, K., Han, Z., Wu, S., Liu, S., and Zou, J.:     |  |  |  |  |  |
| 671 | Decreased N2O and NO emissions associated with stimulated denitrification                       |  |  |  |  |  |
| 672 | following biochar amendment in subtropical tea plantations, Geoderma, 365,                      |  |  |  |  |  |
| 673 | 114223, http://doi.org/10.1016/j.geoderma.2020.114223, 2020.                                    |  |  |  |  |  |
| 674 | Kim, S.Y., Lee, C.H., Gutierrez, J., and Kim, P.J.: Contribution of winter cover crop           |  |  |  |  |  |

| 675 | amendments on global warming potential in rice paddy soil during cultivation,                                                        |
|-----|--------------------------------------------------------------------------------------------------------------------------------------|
| 676 | Plant Soil, 366, 273–286, http://doi.org/10.1007/s11104-012-1403-4, 2013.                                                            |
| 677 | Lang, R., Blagodatsky, S., Xu, J., and Cadisc, G.: Seasonal differences in soil                                                      |
| 678 | respiration and methane uptake in rubber plantation and rainforest, Agric.                                                           |
| 679 | Ecosyst. Environ., 240, 314-328, <a href="http://doi.org/10.1016/j.agee.2017.02.032">http://doi.org/10.1016/j.agee.2017.02.032</a> , |
| 680 | 2017.                                                                                                                                |
| 681 | Lee, L.H., Kim, S.U., Han, H.R., Hur, D., Owens, V.N., Kumar, S., and Hong, C.O.:                                                    |
| 682 | Mitigation of global warming potential and greenhouse gas intensity in arable                                                        |
| 683 | soil with green manure as source of nitrogen, Environ. Pollut., 288, 117724,                                                         |
| 684 | http://doi.org/10.1016/j.envpol.2021.117724, 2021.                                                                                   |
| 685 | Li, B., Gasser, T., Ciais, P., Piao, S., Tao, S., Balkanski, Y., Hauglustaine, D., Boisier,                                          |
| 686 | J.P., Chen, Z., Huang, M., Li, L., Li, Y., Liu, H., Liu, J., Peng, S., Shen, Z., Sun,                                                |
| 687 | Z., Wang, R., Wang, T., Yin, G., Yin, Y., Zeng, H., Zeng, Z., and Zhou, F.: The                                                      |
| 688 | contribution of China's emissions to global climate forcing, Nature, 531, 357-                                                       |
| 689 | 361, http://doi.org/10.1038/nature17165, 2016.                                                                                       |
| 690 | Li, P., Jia, L., Chen, Q., Zhang, H., Deng, J., Lu, J., Xu, L., Li, H., Hu, F., and Jiao, J.:                                        |
| 691 | Adaptive evaluation for agricultural sustainability of different fertilizer                                                          |
| 692 | management options for a green manure-maize rotation system: Impacts on crop                                                         |
| 693 | yield, soil biochemical properties and organic carbon fractions, Sci. Total                                                          |
| 694 | Environ., 908, 168170, http://doi.org/10.1016/j.scitotenv.2023.168170, 2024.                                                         |
| 695 | Lim, S.S., and Choi, W.J.: Changes in microbial biomass, CH <sub>4</sub> and CO <sub>2</sub> emissions, and                          |
| 696 | soil carbon content by fly ash co-applied with organic inputs with contrasting                                                       |
| 697 | substrate quality under changing water regimes, Soil Biol. and Biochem., 68,                                                         |
| 698 | 494–502, http://doi.org/10.1016/j.soilbio.2013.10.027, 2014.                                                                         |
| 699 | Ling, J., Dungait, J.A.J., Delgado, B.M., Cui, Z., Zhou, R., Zhang, W., Gao, Q., Chen,                                               |
| 700 | Y., Yue, S., Kuzyakov, Y., Zhang, F., Chen, X., and Tian, J.: Soil organic carbon                                                    |
| 701 | thresholds control fertilizer effects on carbon accrual in croplands worldwide,                                                      |
| 702 | Nat. Commun., 16, 3009–3009, <a href="http://doi.org/10.1038/s41467-025-57981-6">http://doi.org/10.1038/s41467-025-57981-6</a> ,     |
| 703 | 2025.                                                                                                                                |
| 704 | Liu, S., Lin, F., Wu, S., Ji, C., Sun, Y., Jin, Y., Li, S., Li, Z., and Zou, J.: A meta-analysis                                     |

of fertilizer-induced soil NO and combined NO+N2O emissions. Glob. Change 705 Biol., 23, 2520–2532, http://doi.org/10.1111/gcb.13485, 2016. 706 Liu, Z., Liu, W., Liu, H., Gao, T., Zhao, H., Li, G., Han, H., Li, Z., Lal, R., and Ning, 707 T.: Capture of soil respiration for higher photosynthesis with lower CO<sub>2</sub> 708 emission, J. Clean Prod., 246, 119029, 709 http://doi.org/10.1016/j.jclepro.2019.119029, 2020. 710 Liu, C., Bol, R., Ju, X., Ju, X., Tian, J., and Wu, D.: Trade-offs on carbon and nitrogen 711 712 availability lead to only a minor effect of elevated CO2 on potential denitrification in soil, Soil Biol. Biochem., 176, 108888, 713 http://doi.org/https://doi.org/10.1016/j.soilbio.2022.108888, 2023. 714 Mo, F., Yang, D., Wang, X., Crowther, T.W., Vinay, N., Luo, Z., Yu, K., Sun, S., Zhang, 715 F., Xiong, Y., and Liao, Y.: Nutrient limitation of soil organic carbon stocks 716 Soil Biol. Biochem., 192, 717 under straw return, 109360, http://doi.org/10.1016/j.soilbio.2024.109360, 2024. 718 Muhammad, I., Sainju, U.M., Zhao, F., Khan, A., Ghimire, R., Fu, X., and Wang, J.: 719 Regulation of soil CO2 and N2O emissions by cover crops: A meta-analysis, Soil 720 721 Tillage Res., 192, 103–112, http://doi.org/10.1016/j.still.2019.04.020, 2019. Ni, K., Liao, W., Yi, X., Niu, S., Ma, L., Shi, Y., Zhang, Q., Liu, M., and Ruan, J.: 722 Fertilization status and reduction potential in tea gardens of China, J. Plant Nutr. 723 724 Fert., 25, 421–432, 2019. 725 Pang, J., Li, H., Tang, X., and Geng, J.: Carbon dynamics and environmental controls of a hilly tea plantation in Southeast China, Ecol. Evol. 9, 9723-9735, 726 727 http://doi.org/10.1002/ece3.5504, 2019. Qian, H., Zhu, X., Huang, S., Linquist, B., Kuzyakov, Y., Wassmann, R., Minamikawa, 728 K., Martinez, E.M., Yan, X., Zhou, F., Sander, B.O., Zhang, W., Shang, Z., Zou, 729 J., Zheng, X., Li, G., Liu, Z., Wang, S., Ding, Y., van Groenigen, K.J., and Jiang, 730 731 Y.: Greenhouse gas emissions and mitigation in rice agriculture, Nat. Rev. Earth Environ., 4, 716–732, http://doi.org/10.1038/s43017-023-00482-1, 2023. 732 Schnecker, J., Baldaszti, L., Gündler, P., Pleitner, M., Sandén, T., Simon, E., Spiegel, 733 F., Spiegel, H., Urbina Malo, C., Zechmeister-Boltenstern, S., and Richter, A.: 734

| 735                                                                              | Seasonal dynamics of soil microbial growth, respiration, biomass, and carbon                                                                                                                                                                                                                                                                                                                                                                                                                                                                                                                                                                                                                                                                                                                                                                                                                                                      |  |  |  |  |  |
|----------------------------------------------------------------------------------|-----------------------------------------------------------------------------------------------------------------------------------------------------------------------------------------------------------------------------------------------------------------------------------------------------------------------------------------------------------------------------------------------------------------------------------------------------------------------------------------------------------------------------------------------------------------------------------------------------------------------------------------------------------------------------------------------------------------------------------------------------------------------------------------------------------------------------------------------------------------------------------------------------------------------------------|--|--|--|--|--|
| 736                                                                              | use efficiency in temperate soils, Geoderma 440, 116693,                                                                                                                                                                                                                                                                                                                                                                                                                                                                                                                                                                                                                                                                                                                                                                                                                                                                          |  |  |  |  |  |
| 737                                                                              | http://doi.org/10.1016/j.geoderma.2023.116693, 2023.                                                                                                                                                                                                                                                                                                                                                                                                                                                                                                                                                                                                                                                                                                                                                                                                                                                                              |  |  |  |  |  |
| 738                                                                              | Song, L., Liao, W., Wang, Y., Su, Y., Zhang, Y., Luo, Y., and Sun, L.: Effects of                                                                                                                                                                                                                                                                                                                                                                                                                                                                                                                                                                                                                                                                                                                                                                                                                                                 |  |  |  |  |  |
| 739                                                                              | Interplanting Green Manure on Soil Physico-chemical Characters in Tea                                                                                                                                                                                                                                                                                                                                                                                                                                                                                                                                                                                                                                                                                                                                                                                                                                                             |  |  |  |  |  |
| 740                                                                              | Plantation, Soils 48, 675–679, 2016.                                                                                                                                                                                                                                                                                                                                                                                                                                                                                                                                                                                                                                                                                                                                                                                                                                                                                              |  |  |  |  |  |
| 741                                                                              | Song, J., Song, J., Xu, W., Gao, G., Bai, J., Zhang, Z., Yu, Q., Hao, J., Yang, G., Ren,                                                                                                                                                                                                                                                                                                                                                                                                                                                                                                                                                                                                                                                                                                                                                                                                                                          |  |  |  |  |  |
| 742                                                                              | G., Feng, Y., and Wang, X.: Straw return with fertilizer improves soil CO2                                                                                                                                                                                                                                                                                                                                                                                                                                                                                                                                                                                                                                                                                                                                                                                                                                                        |  |  |  |  |  |
| 743                                                                              | emissions by mitigating microbial nitrogen limitation during the winter wheat                                                                                                                                                                                                                                                                                                                                                                                                                                                                                                                                                                                                                                                                                                                                                                                                                                                     |  |  |  |  |  |
| 744                                                                              | season, Catena 241, 108050, <a href="http://doi.org/10.1016/j.catena.2024.108050">http://doi.org/10.1016/j.catena.2024.108050</a> , 2024.                                                                                                                                                                                                                                                                                                                                                                                                                                                                                                                                                                                                                                                                                                                                                                                         |  |  |  |  |  |
| 745                                                                              | Struck, I.J.A., Taube, F., Hoffmann, M., Kluß, C., Herrmann, A., Loges, R., and Reinsch,                                                                                                                                                                                                                                                                                                                                                                                                                                                                                                                                                                                                                                                                                                                                                                                                                                          |  |  |  |  |  |
| 746                                                                              | T.: Full greenhouse gas balance of silage maize cultivation following grassland:                                                                                                                                                                                                                                                                                                                                                                                                                                                                                                                                                                                                                                                                                                                                                                                                                                                  |  |  |  |  |  |
| 747                                                                              | Are no-tillage practices favorable under highly productive soil conditions?, Soil                                                                                                                                                                                                                                                                                                                                                                                                                                                                                                                                                                                                                                                                                                                                                                                                                                                 |  |  |  |  |  |
| 748                                                                              | Tillage Res., 200, 104615, <a href="http://doi.org/10.1016/j.still.2020.104615">http://doi.org/10.1016/j.still.2020.104615</a> , 2020.                                                                                                                                                                                                                                                                                                                                                                                                                                                                                                                                                                                                                                                                                                                                                                                            |  |  |  |  |  |
| 749                                                                              | Tang, J., Baldocchi, D.D., and Xu, L.: Tree photosynthesis modulates soil respiration                                                                                                                                                                                                                                                                                                                                                                                                                                                                                                                                                                                                                                                                                                                                                                                                                                             |  |  |  |  |  |
| 750                                                                              | on a diurnal time scale, Glob. Change Biol., 11, 1298-1304,                                                                                                                                                                                                                                                                                                                                                                                                                                                                                                                                                                                                                                                                                                                                                                                                                                                                       |  |  |  |  |  |
| 750                                                                              | on a diamai time scale, Glob. Change Blot., 11, 1276–1304,                                                                                                                                                                                                                                                                                                                                                                                                                                                                                                                                                                                                                                                                                                                                                                                                                                                                        |  |  |  |  |  |
| 751                                                                              | http://doi.org/10.1111/j.1365-2486.2005.00978.x, 2005.                                                                                                                                                                                                                                                                                                                                                                                                                                                                                                                                                                                                                                                                                                                                                                                                                                                                            |  |  |  |  |  |
|                                                                                  |                                                                                                                                                                                                                                                                                                                                                                                                                                                                                                                                                                                                                                                                                                                                                                                                                                                                                                                                   |  |  |  |  |  |
| 751                                                                              | http://doi.org/10.1111/j.1365-2486.2005.00978.x, 2005.                                                                                                                                                                                                                                                                                                                                                                                                                                                                                                                                                                                                                                                                                                                                                                                                                                                                            |  |  |  |  |  |
| 751<br>752                                                                       | http://doi.org/10.1111/j.1365-2486.2005.00978.x, 2005.  Wang, J., Smith, P., Hergoualc'h, K., and Zou, J.: Direct N <sub>2</sub> O emissions from global tea                                                                                                                                                                                                                                                                                                                                                                                                                                                                                                                                                                                                                                                                                                                                                                      |  |  |  |  |  |
| 751<br>752<br>753                                                                | http://doi.org/10.1111/j.1365-2486.2005.00978.x, 2005.  Wang, J., Smith, P., Hergoualc'h, K., and Zou, J.: Direct N <sub>2</sub> O emissions from global tea plantations and mitigation potential by climate-smart practices, Resour. Conserv.                                                                                                                                                                                                                                                                                                                                                                                                                                                                                                                                                                                                                                                                                    |  |  |  |  |  |
| 751<br>752<br>753<br>754                                                         | http://doi.org/10.1111/j.1365-2486.2005.00978.x, 2005.  Wang, J., Smith, P., Hergoualc'h, K., and Zou, J.: Direct N <sub>2</sub> O emissions from global tea plantations and mitigation potential by climate-smart practices, Resour. Conserv. Recycl., 185, 106501, <a href="http://doi.org/10.1016/j.resconrec.2022.106501">http://doi.org/10.1016/j.resconrec.2022.106501</a> , 2022.                                                                                                                                                                                                                                                                                                                                                                                                                                                                                                                                          |  |  |  |  |  |
| 751<br>752<br>753<br>754<br>755                                                  | http://doi.org/10.1111/j.1365-2486.2005.00978.x, 2005.  Wang, J., Smith, P., Hergoualc'h, K., and Zou, J.: Direct N <sub>2</sub> O emissions from global tea plantations and mitigation potential by climate-smart practices, Resour. Conserv. Recycl., 185, 106501, <a href="http://doi.org/10.1016/j.resconrec.2022.106501">http://doi.org/10.1016/j.resconrec.2022.106501</a> , 2022.  Wang, B., Wang, S., Li, G., Fu, L., Chen, H., Yin, M., and Chen, J.: Reducing nitrogen                                                                                                                                                                                                                                                                                                                                                                                                                                                  |  |  |  |  |  |
| 751<br>752<br>753<br>754<br>755<br>756                                           | http://doi.org/10.1111/j.1365-2486.2005.00978.x, 2005.  Wang, J., Smith, P., Hergoualc'h, K., and Zou, J.: Direct N <sub>2</sub> O emissions from global tea plantations and mitigation potential by climate-smart practices, Resour. Conserv. Recycl., 185, 106501, <a href="http://doi.org/10.1016/j.resconrec.2022.106501">http://doi.org/10.1016/j.resconrec.2022.106501</a> , 2022.  Wang, B., Wang, S., Li, G., Fu, L., Chen, H., Yin, M., and Chen, J.: Reducing nitrogen fertilizer usage coupled with organic substitution improves soil quality and                                                                                                                                                                                                                                                                                                                                                                     |  |  |  |  |  |
| 751<br>752<br>753<br>754<br>755<br>756<br>757                                    | http://doi.org/10.1111/j.1365-2486.2005.00978.x, 2005.  Wang, J., Smith, P., Hergoualc'h, K., and Zou, J.: Direct N <sub>2</sub> O emissions from global tea plantations and mitigation potential by climate-smart practices, Resour. Conserv. Recycl., 185, 106501, <a href="http://doi.org/10.1016/j.resconrec.2022.106501">http://doi.org/10.1016/j.resconrec.2022.106501</a> , 2022.  Wang, B., Wang, S., Li, G., Fu, L., Chen, H., Yin, M., and Chen, J.: Reducing nitrogen fertilizer usage coupled with organic substitution improves soil quality and boosts tea yield and quality in tea plantations, J. Sci. Food Agric., 105, 1228–                                                                                                                                                                                                                                                                                    |  |  |  |  |  |
| 751<br>752<br>753<br>754<br>755<br>756<br>757<br>758                             | http://doi.org/10.1111/j.1365-2486.2005.00978.x, 2005.  Wang, J., Smith, P., Hergoualc'h, K., and Zou, J.: Direct N <sub>2</sub> O emissions from global tea plantations and mitigation potential by climate-smart practices, Resour. Conserv. Recycl., 185, 106501, <a href="http://doi.org/10.1016/j.resconrec.2022.106501">http://doi.org/10.1016/j.resconrec.2022.106501</a> , 2022.  Wang, B., Wang, S., Li, G., Fu, L., Chen, H., Yin, M., and Chen, J.: Reducing nitrogen fertilizer usage coupled with organic substitution improves soil quality and boosts tea yield and quality in tea plantations, J. Sci. Food Agric., 105, 1228–1238, <a href="http://doi.org/10.1002/jsfa.13913">http://doi.org/10.1002/jsfa.13913</a> , 2024.                                                                                                                                                                                     |  |  |  |  |  |
| 751<br>752<br>753<br>754<br>755<br>756<br>757<br>758<br>759                      | <ul> <li>http://doi.org/10.1111/j.1365-2486.2005.00978.x, 2005.</li> <li>Wang, J., Smith, P., Hergoualc'h, K., and Zou, J.: Direct N<sub>2</sub>O emissions from global tea plantations and mitigation potential by climate-smart practices, Resour. Conserv. Recycl., 185, 106501, <a href="http://doi.org/10.1016/j.resconrec.2022.106501">http://doi.org/10.1016/j.resconrec.2022.106501</a>, 2022.</li> <li>Wang, B., Wang, S., Li, G., Fu, L., Chen, H., Yin, M., and Chen, J.: Reducing nitrogen fertilizer usage coupled with organic substitution improves soil quality and boosts tea yield and quality in tea plantations, J. Sci. Food Agric., 105, 1228–1238, <a href="http://doi.org/10.1002/jsfa.13913">http://doi.org/10.1002/jsfa.13913</a>, 2024.</li> <li>Wang, G., Zhao, M., Zhao, B., Liu, X., and Wang, Y.: Reshaping Agriculture Eco-</li> </ul>                                                            |  |  |  |  |  |
| 751<br>752<br>753<br>754<br>755<br>756<br>757<br>758<br>759<br>760               | <ul> <li>http://doi.org/10.1111/j.1365-2486.2005.00978.x, 2005.</li> <li>Wang, J., Smith, P., Hergoualc'h, K., and Zou, J.: Direct №O emissions from global tea plantations and mitigation potential by climate-smart practices, Resour. Conserv. Recycl., 185, 106501, <a href="http://doi.org/10.1016/j.resconrec.2022.106501">http://doi.org/10.1016/j.resconrec.2022.106501</a>, 2022.</li> <li>Wang, B., Wang, S., Li, G., Fu, L., Chen, H., Yin, M., and Chen, J.: Reducing nitrogen fertilizer usage coupled with organic substitution improves soil quality and boosts tea yield and quality in tea plantations, J. Sci. Food Agric., 105, 1228–1238, <a href="http://doi.org/10.1002/jsfa.13913">http://doi.org/10.1002/jsfa.13913</a>, 2024.</li> <li>Wang, G., Zhao, M., Zhao, B., Liu, X., and Wang, Y.: Reshaping Agriculture Ecoefficiency in China: From Greenhouse Gas Perspective, Ecol. Indic., 172,</li> </ul> |  |  |  |  |  |
| 751<br>752<br>753<br>754<br>755<br>756<br>757<br>758<br>759<br>760<br>761        | http://doi.org/10.1111/j.1365-2486.2005.00978.x, 2005.  Wang, J., Smith, P., Hergoualc'h, K., and Zou, J.: Direct N2O emissions from global tea plantations and mitigation potential by climate-smart practices, Resour. Conserv. Recycl., 185, 106501, http://doi.org/10.1016/j.resconrec.2022.106501, 2022.  Wang, B., Wang, S., Li, G., Fu, L., Chen, H., Yin, M., and Chen, J.: Reducing nitrogen fertilizer usage coupled with organic substitution improves soil quality and boosts tea yield and quality in tea plantations, J. Sci. Food Agric., 105, 1228–1238, http://doi.org/10.1002/jsfa.13913, 2024.  Wang, G., Zhao, M., Zhao, B., Liu, X., and Wang, Y.: Reshaping Agriculture Ecoefficiency in China: From Greenhouse Gas Perspective, Ecol. Indic., 172, 113268, http://doi.org/10.1016/j.ecolind.2025.113268, 2025.                                                                                             |  |  |  |  |  |
| 751<br>752<br>753<br>754<br>755<br>756<br>757<br>758<br>759<br>760<br>761<br>762 | http://doi.org/10.1111/j.1365-2486.2005.00978.x, 2005.  Wang, J., Smith, P., Hergoualc'h, K., and Zou, J.: Direct N2O emissions from global tea plantations and mitigation potential by climate-smart practices, Resour. Conserv. Recycl., 185, 106501, http://doi.org/10.1016/j.resconrec.2022.106501, 2022.  Wang, B., Wang, S., Li, G., Fu, L., Chen, H., Yin, M., and Chen, J.: Reducing nitrogen fertilizer usage coupled with organic substitution improves soil quality and boosts tea yield and quality in tea plantations, J. Sci. Food Agric., 105, 1228–1238, http://doi.org/10.1002/jsfa.13913, 2024.  Wang, G., Zhao, M., Zhao, B., Liu, X., and Wang, Y.: Reshaping Agriculture Ecoefficiency in China: From Greenhouse Gas Perspective, Ecol. Indic., 172, 113268, http://doi.org/10.1016/j.ecolind.2025.113268, 2025.  Wanyama, I., Pelster, D.E., Butterbach-Bahl, K., Verchot, L. V., Martius, C., and Rufino,  |  |  |  |  |  |

http://doi.org/10.1007/s10533-019-00555-8, 2019. Widanagamage, N., Santos, E., Rice, C.W., and Patrignani, A.: Study of soil 766 heterotrophic respiration as a function of soil moisture under different land 767 covers, Soil Biol. Biochem., 200, 109593, 768 http://doi.org/10.1016/j.soilbio.2024.109593, 2025. 769 WMO: Greenhouse Gas Bulletin-No. 20: The state of greenhouse gases in the 770 atmosphere based on global observations through 2023, WMO Greenhouse Gas 771 Bulletin, Switzerland, 2024. 772 Wu, Y., Li, Y., Fu, X., Shen, J., Chen, D., Wang, Y., Liu, X., Xiao, R., Wei, W., and Wu, 773 J.: Effect of controlled-release fertilizer on N2O emissions and tea yield from a 774 tea field in subtropical central China, Environ. Sci. Pollut. Res., 25. 25580-775 25590, http://doi.org/10.1007/s11356-018-2646-2, 2018. 776 Wu, Y., Li, Y., Wang, H., Wang, Z., Fu, X., Shen, J., Wang, Y., Liu, X., Meng, L., and 777 778 Wu, J.: Response of N<sub>2</sub>O emissions to biochar amendment on a tea field soil in subtropical central China: A three-year field experiment, Agric. Ecosyst. 779 Environ., 318, 107473, http://doi.org/10.1016/j.agee.2021.107473, 2021. 780 781 Xin, W., Zhang, J., Yu, Y., Tian, Y., Li, H., Chen, X., Li, W., Liu, Y., Lu, T., He, B., 782 Xiong, Y., Yang, Z., Xu, T., and Tang, W.: Root microbiota of tea plants regulate 783 nitrogen homeostasis and theanine synthesis to influence tea quality, Curr. Biol., 784 34, 868–880, http://doi.org/10.1016/j.cub.2024.01.044, 2024. Xu, B., and Lin, B.: Factors affecting CO<sub>2</sub> emissions in China's agriculture sector: 785 Evidence from geographically weighted regression model, Energy Policy, 104, 786 787 404–414. <a href="http://doi.org/10.1016/j.enpol.2017.02.011">http://doi.org/10.1016/j.enpol.2017.02.011</a>, 2017. Xu, X., Zhao, Q., Guo, J., Li, C., Li, J., Niu, K., Jin, S., Fu, C., Gaffney, P.P.J., Xu, Y., 788 Sun, M., Xue, Y., Chang, D., Zhang, Y., Si, W., Fan, S., and Zhang, L.: 789 Inequality in agricultural greenhouse gas emissions intensity has risen in rural 790 1993 791 China from 2020, Nat. Food, 5, 916-928, http://doi.org/10.1038/s43016-024-01071-1, 2024. 792 Yan, P., Wu, L., Wang, D., Fu, J., Shen, C., Li, X., Zhang, L., Zhang, L., Fan, L., and 793 Han W.: Soil acidification in Chinese tea plantations, Sci. Total Environ., 715, 794

| 795 | 136963, http://doi.org/10.1016/j.scitotenv.2020.136963, 2020.                                                                      |
|-----|------------------------------------------------------------------------------------------------------------------------------------|
| 796 | Yan, W., Zhong, Y., Yang, J., and Torn, M.S.: Response of soil greenhouse gas fluxes                                               |
| 797 | to warming: A global meta-analysis of field studies, Geoderma, 419, 115865,                                                        |
| 798 | http://doi.org/10.1016/j.geoderma.2022.115865, 2022.                                                                               |
| 799 | Yang, X., Xiong, J., Du, T., Ju, X., Gan, Y., Li, S., Xia, L., Shen, Y., Pacenka, S.,                                              |
| 800 | Steenhuis, T.S., Siddique, K.H.M., Kang, S., and Butterbach-Bahl, K.:                                                              |
| 801 | Diversifying crop rotation increases food production, reduces net greenhouse                                                       |
| 802 | gas emissions and improves soil health, Nat. Commun., 15, 198,                                                                     |
| 803 | http://doi.org/10.1038/s41467-023-44464-9, 2024.                                                                                   |
| 804 | Yao, Z., Wei, Y., Liu, C., Zheng, X., Xie, B.: Organically fertilized tea plantation                                               |
| 805 | stimulates N2O emissions and lowers NO fluxes in subtropical China,                                                                |
| 806 | Biogeosciences, 12, 5915-5928, <a href="http://doi.org/10.5194/bg-12-5915-2015">http://doi.org/10.5194/bg-12-5915-2015</a> , 2015. |
| 807 | Zhang, Q., Lei, H., Yang, D., Xiong, L., Liu, P., and Fang, B.: Decadal variation in CO <sub>2</sub>                               |
| 808 | fluxes and its budget in a wheat and maize rotation cropland over the North                                                        |
| 809 | China Plain, Biogeosciences, 17, 2245-2262, <a href="http://doi.org/10.5194/bg-17-">http://doi.org/10.5194/bg-17-</a>              |
| 810 | <u>2245-2020,</u> 2020.                                                                                                            |
| 811 | Zhang, W., Lu, J., Bai, J., Khan, A., Liu, S., Zhao, L., Wang, W., Zhu, S., Li, X., Tian,                                          |
| 812 | X., Li, S., and Xiong, Y.: Introduction of soybean into maize field reduces N2O                                                    |
| 813 | emission intensity via optimizing nitrogen source utilization, J. Clean Prod., 442,                                                |
| 814 | 141052, http://doi.org/10.1016/j.jclepro.2024.141052, 2024.                                                                        |
| 815 | Zhu, Z., Ge, T., Luo, Y., Liu, S., Xu, X., Tong, C., and Shibistova, O., Guggenberger,                                             |
| 816 | G., Wu, J.: Microbial stoichiometric flexibility regulates rice straw                                                              |
| 817 | mineralization and its priming effect in paddy soil, Soil Biol. Biochem., 121,                                                     |
| 818 | 67–76, http://doi.org/10.1016/j.soilbio.2018.03.003, 2018.                                                                         |
| 819 | Zou, J., Huang, Y., Qin, Y., Shen, Q., Pan, G., Lu, Y., and Liu, Q.: Changes in fertilizer-                                        |
| 820 | induced direct N <sub>2</sub> O emissions from paddy fields during rice-growing season in                                          |
| 821 | China between 1950s and 1990s, Glob. Change Biol., 15, 229–242,                                                                    |
| 822 | http://doi.org/10.1111/j.1365-2486.2008.01775.x, 2009.                                                                             |

# **Tables and Figures**

**Table 1.** Seasonal variation in  $CO_2$  fluxes from tea rows and inter-row zones under different green manure intercropping treatments. Different lowercase letters indicate significant differences among treatments and seasons (P < 0.05).

| Туре | Spring                             | Summer                             | Autumn                             | Winter                             |  |
|------|------------------------------------|------------------------------------|------------------------------------|------------------------------------|--|
|      | $(mg \cdot m^{-2} \cdot min^{-1})$ |  |
| CKT  | $6.46 \pm 0.58^{bcd}$              | 9.66±0.49 <sup>a</sup>             | 9.93±0.68 <sup>a</sup>             | 3.60±0.36 <sup>f</sup>             |  |
| HMT  | $5.65{\pm}0.44^{def}$              | $7.97 \pm 0.39^{abc}$              | $9.21 \pm 0.46^{a}$                | $4.10{\pm}0.43^{ef}$               |  |
| SMT  | $6.05{\pm}0.39^{cde}$              | $8.99{\pm}0.34^{a}$                | $8.47{\pm}0.30^{ab}$               | $3.71 \pm 0.37^{\rm f}$            |  |
| CKG  | $8.83{\pm}1.14^{cd}$               | $10.63{\pm}0.66^{abc}$             | $9.76{\pm}0.83^{ab}$               | $4.03\pm0.52^{e}$                  |  |
| HMG  | $9.08 \pm 0.49^{cd}$               | $13.03 \pm 0.29^a$                 | $9.38 \pm 0.39^{bcd}$              | 4.19±0.29e                         |  |
| SMG  | $10.76 \pm 0.43^{ab}$              | $12.69\pm0.73^{ab}$                | $10.94 \pm 0.86^{abc}$             | $6.03\pm0.84^{de}$                 |  |

828

824825

826

832

**Figure 1.** (a) Geographic location of the study area in Shengzhou City, Zhejiang Province, China; (b) field layout of the tea plantation experiment; (c) photos of *Lolium perenne* L. and *Trifolium repens* L. plot, and *Vulpia myuros* C. plot, the bare control plot, respectively.

835

836

**Figure 2.** Dynamics of (a) air temperature and precipitation, (b) CO<sub>2</sub> fluxes from tea rows, and (c) CO<sub>2</sub> fluxes from inter-rows during the observation period (2022–2024). Black, green and orange arrows represent the timings of fertilization, grass planting and tea pruning, respectively.

840

841842

**Figure 3.** (a, b) Annual cumulative CO<sub>2</sub> emissions from tea rows and inter-rows under different green manure intercropping treatments; (c, d) contribution of tea rows and inter-rows to total annual CO<sub>2</sub> emissions under each treatment.

844

845

**Figure 4.** Diurnal variation in  $CO_2$  fluxes from (a, c, e, g) tea rows and (b, d, f, h) interrow zones under different green manure intercropping treatments across seasons: (a-b) spring, (c-d) summer, (e-f) autumn, and (g-h) winter.

848

849

850

**Figure 5.** Temporal dynamics of CO<sub>2</sub> fluxes under green manure (a, b) growth stages and (c, d) management events in tea plantations. Growth stages include: early growth (mid-November to early April), vigorous growth (mid-April to late May), wilting (early June to late July), and decomposition (August).

854

**Figure 6.** Basic physicochemical properties of soil in tea rows and inter-rows under different green manure intercropping treatments.

856

**Figure 7.** Pairwise correlations between environmental factors and their relationships with CO<sub>2</sub> fluxes under different green manure treatments (\*p 

**Figure 8.** Canonical correspondence analysis (CCA) showing the influence of soil physicochemical properties on CO<sub>2</sub> emissions from tea rows and inter-rows under different green manure treatments.