# Peer review of "Spatially Contrasting CO2 Dynamics Driven by Green Manure"

_EGUsphere, 2025_

## Referee Comment (RC2)

**General Comments**

This study focuses on a subtropical tea plantation and utilizes the static chamber–gas chromatography method to conduct long-term monitoring of $CO_2$ emission fluxes under different green manure intercropping treatments. It systematically reveals the short- and long-term trends and differences in $CO_2$ fluxes across different areas of the tea garden, including between tea rows, and analyzes the impact of key environmental drivers such as soil temperature, moisture, and organic carbon on carbon emissions. The findings provide important insights into the carbon cycling mechanisms in tea plantations under green manure intercropping and hold significant practical value for promoting low-carbon management in tea cultivation. The experimental design is reasonable, the analysis and discussion are thorough, and the structure is clear, making the paper suitable for publication in this journal. The following minor revisions are recommended:

1. Lines 153–154: The baseline physicochemical properties of the soil in the study area are not provided. It is recommended to include initial soil characteristics.
2. Line 171: In addition to the green manure treatment, please specify the type and application rate of fertilizers used.
3. Line 185: For consistency, please express time in 24-hour format. Replace "between 9:00 and 11:00 a.m." with "between 09:00 and 11:00 (local time)."
4. Line 198: What was the soil sampling depth? This significantly influences soil physicochemical properties. Sampling across multiple soil layers would be more appropriate.
5. Lines 216–217: The method for calculating $CO_2$ fluxes is not clearly described. Please provide details.
6. Lines 308–310: On what basis were the growth stages of green manure defined? Please clarify.
7. Lines 498–499: The decrease in cumulative emissions between rows in the second year is attributed to "reduced human disturbance," which is insufficiently supported. Please elaborate with references to relevant literature.
8. Line 541: The discussion on the SOC threshold lacks adequate references. Additional literature should be cited and discussed.
9. Table 1: Please clearly note "Values are mean ± SE" in the table caption or footnote.
10. Root biomass data are lacking. The contribution of green manure roots to soil respiration has not been quantified, which may affect the interpretation of $CO_2$ flux sources.
11. Within the closed chamber environment, temperature and humidity change over time, potentially influencing $CO_2$ flux measurements. Further analysis on this aspect is recommended.
12. The study does not explore how different green manure treatments regulate microbial activity or the mechanisms by which soil microbial communities drive carbon sequestration. If relevant measurements were not included, this could be addressed in future research.
13. The language throughout the manuscript should be further refined to avoid repetitive statements, particularly in the Results and Discussion sections.

---

## Author Comment (AC1)

**Revision EGUSPHERE-2025-5065**

**Point by point response to the reviewers:**

First of all, we would like to thank the editor and the reviewers for the positive and constructive comments. We carefully revised and improved the article. The followings are response to the comments of reviewers. The line number below is indicated based on the **clean version.**

**General Comments**

The manuscript presents valuable insights from a two-year measurement campaign of $CO_2$ fluxes in a tea plantation in China. The study compares plots subjected to different intercropping treatments, providing an interesting perspective by distinguishing emissions from tea rows and inter-row spaces. The discussion is comprehensive; however, additional experimental details would enhance the reader's ability to interpret the results. Aside from this, I have only a few minor comments (see below). Overall, the paper falls within the scope of *Biogeosciences* and merits publication after these minor revisions are addressed.

**Response:** Thank you very much for your positive comments. We have revised the manuscript and answered the questions point by point. The line number below is indicated based on the clean version.

**Specific Comments:**

**Abstract**

1. Lines 28–31: These absolute values are difficult to interpret without broader context regarding tea plantation emissions. Readers will likely be more interested in relative changes compared to control plots, as mentioned in line 41. Please provide relative changes for the total flux as well (see comment on line 41).

   **Response:** Thank you for the comments Your viewpoint is indeed more accurate. We have removed the average values and replaced them with the relative changes in average fluxes. We revised the description (Lines 29–32).

   "Average tea-row fluxes were 8.7% and 9.5% lower under SM and HM,

respectively, compared to CK, indicating emission reductions with intercropping. In contrast, average inter-row fluxes increased by 19.4% under SM and 7.7% under HM, demonstrating pronounced spatial contrasts."

2. Lines 32–33: Specify for which plots midday peaks occurred.

**Response:** Thank you for the comments. While diurnal variations differed slightly between tea rows and inter-rows and among different treatments, it is worth noting that, overall, the peak fluxes predominantly occurred around midday. We revised the description (Lines 32–35).

"Diurnal patterns generally exhibited midday peaks (12:00–14:00), especially in summer and autumn across all tea-rows, and short-term $CO_2$ pulses were triggered by field operations such as fertilization and pruning."

3. Line 41: The relative changes reported do not correspond to the values in lines 28–31. Be more precise which aggregates are shown and used for the relative changes.

**Response:** Thank you for the comments. The reported percentage changes are based on the total annual emissions (derived from the fluxes), rather than on the average flux values alone, and have been revised accordingly as suggested. We revised the description (Lines 41–43).

"Compared to CK, intercropping reduced tea-row emissions by 7.1–7.9% but increased inter-row emissions by 12.7–28.9% based on the two years cumulative emissions, continuous intercropping significantly decreased overall inter-row emissions over time."

4. Lines 40–45: This section is unclear. Tea-row and inter-row fluxes appear similar in magnitude (e.g., 8.12 CK (tea-row) vs. 9.07 CK (inter-row)). Tea-row emissions decrease by 7–8%, while inter-row emissions increase by 13–29%. How does this support the conclusion that intercropping is a climate-smart management strategy?

**Response:** Thank you for the question. This question is of great significance. In fact, the comparisons in the text should be understood within the correct frame of reference. Whether for the tea rows or the inter-rows, the emission data are relative

to "tea plantations with no fertilization." For traditional agricultural economic tea plantations, the amount of fertilizer applied is huge. Most importantly, the overall emissions from the "green manure intercropping" tea plantation remain significantly lower than those from conventionally fertilized tea plantations. This demonstrates that intercropping, as an integrated system, has a clear emission reduction effect.

Furthermore, and more importantly, evidence on the temporal scale shows that over the two-year period, emissions from the tea rows continued to decrease, while emissions from the inter-rows dropped significantly in the second year compared to the first. This powerfully indicates that the emission reduction benefits of green manure intercropping require time to stabilize and fully manifest across the entire system, including the inter-rows, highlighting the critical importance of sustained implementation for achieving long-term and stable emission reductions.

5. Add to the abstract: measurement method (flux chambers), campaign duration, and dataset size.

**Response:** Thank you for the comments. We revised the description (Lines 23–27).

"We employed the static chamber method over a two-year period, with sampling conducted weekly, to investigate how intercropping with *Vulpia myuros* (SM) and a legume–nonlegume mixture of *Lolium perenne* and *Trifolium repens* (HM) influenced spatial $CO_2$ flux dynamics compared with a no-intercropping control (CK) from tea rows and inter-row zones in a subtropical tea plantation."

**Experimental Setup (Lines 164 ff.)**

1. Provide more details: plantation size, distances between plots and replicates, age of plantation and tea plants, etc.

**Response:** Thank you for your valuable and professional comments. The detailed information has been added to Section 2.1 (Lines 161–164) and Section 2.2 (Lines 171–175), respectively.

The added content is as follows:

In Section 2.1 (Lines 161–164):

"The tea plantation was established in 2015; the tea plants are 8–10 years old and arranged in a single-row planting pattern with a row spacing of 150 cm and a plant spacing of 40 cm."

In Section 2.2 (Lines 171–175):

"The experimental tea plantation covered an area of approximately 1000 m². For each treatment, three representative tea rows were selected as three replicates. Adjacent treatments were separated by two tea rows (~3 m), and replicate areas within the same treatment were spaced approximately 5 m apart."

2. Consider adding a schematic (with dimensions) to Figure 1. Clarify whether tea plants were inside the chambers.

**Response:** Thank you for your constructive comment. Figure 1b is a field photo of the experimental setup. The silver chamber shown is the flux chamber, which fully enclosed the tea plant. We have revised the figure by adding the specific dimensions of the flux chamber next to it in the image. (A photo of the chamber placed in an inter-rows location has also been added to panel b.)

[Figure]

**Figure 1.** (a) Geographic location of the study area in Shengzhou City, Zhejiang

Province, China; (b) field layout of the tea plantation experiment; (c) photos of *Lolium perenne* L. and *Trifolium repens* L. plot, and *Vulpia myuros* C. plot, the blank control plot, respectively.

3. According to lines 183–184, one measurement included four samples every 7 minutes: indicate when the first sample was taken after chamber closure and how long (overall) a single experiment took. How high were the $CO_2$ concentrations at the end?

**Response:** Thank you for your questions. This comment covers several issues; we have addressed each of the points raised below in a point-by-point manner:

We collected four gas samples during the 21-minute period after chamber closure to calculate the flux at each point. The $CO_2$ concentration of the final gas sample is provided in the representative data table below. Data shown are means ± SE.

| Type | Spring (ppm) | Summer (ppm) | Autumn (ppm) | Winter (ppm) |
|------|------|------|------|------|
| CKT | 559.38±9.04 | 603.10±27.82 | 586.91±10.10 | 506.64±5.33f |
| HMT | 550.06±9.72 | 569.38±23.78 | 574.68±8.07 | 507.70±4.94 |
| SMT | 550.44±8.61 | 583.63±24.70 | 575.51±8.85 | 503.52±4.85 |
| CKG | 705.94±26.25 | 705.93±30.60 | 662.62±16.00 | 563.34±6.88 |
| HMG | 719.34±24.49 | 725.38±34.82 | 681.12±15.92 | 559.16±10.55 |
| SMG | 740.50±27.08 | 757.70±36.72 | 708.70±19.18 | 581.16±11.75 |

Include an example showing the 4 measured $CO_2$ concentrations over time (perhaps in the appendix) and illustrate how fluxes were calculated. Was a linear fit applied to the four data points? How linear was the $CO_2$ increase? How consistent were the three replicates?

**Response:** Thank you for your questions. Fluxes were calculated by determining the linear slope from four gas concentration measurements. We added the formula in Appendix 1. All fitted slopes selected for flux calculation exhibited high linearity,

with $R^2 > 0.95$. To ensure data reliability, outliers and points with large standard deviations were excluded during quality control. The resulting consistency across replicates is reflected in the standard deviations of the mean fluxes, as shown in Fig. 2b, c. The following figure illustrates the slope of the linear regression of $CO_2$ concentration over time.

[Figure]

Were all six plots always sampled simultaneously? Did you ever observe decreases in $CO_2$ concentrations? Figure 2 shows only positive fluxes.

**Response:** Thank you for your questions. Sampling was conducted sequentially

across the six treatments, and each full sampling cycle was typically completed within one hour. Dark chambers covered with aluminum foil were used to measure respiratory fluxes. No significant decline in $CO_2$ concentration was observed during long-term flux measurements. However, a limited number of negative flux values were recorded during diurnal observations in spring and winter, as detailed in the diurnal variation section.

Include an uncertainty analysis for flux determination.

**Response:** Thank you for your comments. We added it (Lines 201–208)

"During the tests, the deviation between the calculated regression values of $CO_2$ and the nominal mole fractions was 0.37 μmol mol$^{-1}$. The linear fit between the instrument response values and the nominal mole fractions achieved a correlation coefficient ($R^2$) of 0.9999. Furthermore, the standard gases used were calibrated in multiple rounds by the Greenhouse Gas Laboratory of the Atmospheric Observation Center of the China Meteorological Administration using primary standard gases, ensuring traceability to the World Meteorological Organization primary standards."

**Figure and Table Captions**

1. Add a condensed explanation of your three letter codes to the different figure/table captions: Something like: CK = control, SM and HM = intercropping types, T = tea row, G = inter-row.

   **Response:** Thank you for this helpful suggestion. We agree that adding a condensed explanation of the letter codes to the figure and table captions will significantly improve their clarity for readers. Accordingly, we add the following note to all relevant captions: "CK for control; SM and HM for intercropping types, T for tea row, G for inter-row."

**Results**

1.  Line 243: Explain what the uncertainties represent.

    **Response:** The uncertainty mentioned here refers to the standard deviation. We have added a clarification regarding the flux data presentation in the text (Line 239) as follows: "Data shown are means ± standard error (SE)."

2.  Line 245: Specify the significance level for "significantly higher."

    **Response:** Thank you for your suggestion. We have added the notation "($p < 0.05$)" in the "2.3 Data Processing" section (Lines 239–240).

    "In all statistical tests, the level of significant differences and correlations was set at $p < 0.05$."

**Figures and Tables**

1.  Figure 2: Clarify whether error bars indicate uncertainty of a single flux measurement or standard deviation across duplicates.

    **Response:** Thank you for your comments. The error bars represent the standard error of the mean among repeated measurements. We have added the relevant description in the Figure 2.

2.  Table 1: Explain ± values and superscript letters.

    **Response:** Thank you for your comments. In this study, data are presented as mean ± standard error (SE). The use of SE is preferred when making inferences about the population mean or comparing the differences between group means. The superscript letters indicate significant differences between seasons and treatment types, as determined by two-way ANOVA. Different letters denote statistically significant differences ($p < 0.05$). we added the descriptions in Table 1.

3.  Figure 2: Use a different color for precipitation vs. temperature. Confirm whether

negative CO₂ fluxes were ever observed. Were tea plants present in the chambers?

**Response:** Thank you for your comments. We have revised Figure 2. Measurements were predominantly conducted on clear mornings, with net $CO_2$ emissions observed in the majority of cases and only weak negative fluxes recorded at a few time points. The tea plant was enclosed inside the measurement chamber during sampling; it can be seen in revised Figure 1b.

[Figure]

4. Figure 3: Explain superscript letters and error bars. Add "T" for tea row and "G" for inter-row in the legend.

**Response:** Thank you for your comments. In this study, data are presented as mean ± standard error; superscript letters denote significant groupings based on two-way ANOVA with Tukey's post-hoc test. We added it (Lines 892–893).

"Data shown are means ± SE. Different superscript letters denote statistically significant differences ($p < 0.05$). CK for control; SM and HM for intercropping types, T for tea row, G for inter-row." in the Figure 3 legend.

**Results**

1. Lines 354 ff.: The consistent differences in soil properties are surprising given the short distances. Provide more detail on plantation history and intercropping duration. When did the intercropping practices start?

   **Response:** Thank you for your comments. The CKG and SMG treatments were spatially separated by three tea rows (approximately 5 m). Green manure intercropping was maintained for two years, with sowing conducted each November and decomposition occurring by August of the following year. Under the SM treatment, which involved continuous coverage of *Vulpia myuros* over two years, hence changes in soil properties were observed. The presented data represent the two-year averages of soil ammonium-N and nitrate-N concentrations. We added more detail on plantation history and intercropping duration. (Lines 161–164; 171–174; 178–180)

   "The tea plantation was established in 2015; the tea plants are 8–10 years old and arranged in a single-row planting pattern with a row spacing of 150 cm and a plant spacing of 40 cm."

   "The experimental tea plantation covered an area of approximately 1000 m². For each treatment, three representative tea rows were selected as three replicates. Adjacent treatments were separated by two tea rows (~3 m), and replicate areas within the same treatment were spaced approximately 5 m apart."

   "Basal fertilization and tillage were conducted in middle of October, followed by green manure sowing in early November."

**Conclusions**

1. Line 587 refers to the mitigation of carbon losses. Quantify overall $CO_2$ emission reductions when summing tea-row and inter-row fluxes for HM, SM, and CK. Data (e.g. in abstract, see my comment there) do not support it and Figure 3 suggest reductions only for HM in the second year. If mitigation exists, contextualize its

potential impact at larger scales.

**Response:** Thank you for your comments. The values mentioned are relative to the CK treatment, and we have added "compared to the CK" in line 602. We also recognize that the net mitigation effect at the small plot scale over a two-year period may be limited. The mitigation effect is primarily manifested as a significant reduction in emissions from the tea-row zone, which offsets the initial increase observed in the inter-row area. The clear declining trend in inter-row emissions under the HM treatment in the second year suggests that the initial emission pulse was likely a transient response to soil disturbance during the green-manure establishment phase, and that its mitigation potential strengthens over time. Future management could be optimized to maximize emission reduction in tea rows while minimizing disturbance in inter-row areas, thereby enhancing the overall net benefit.

2. Add general considerations on intercropping: benefits and drawbacks (costs, labor, temperature effects, soil compaction, aeration, etc.). Large-scale adoption depends on whether benefits outweigh additional costs.

    **Response:** We thank the reviewer for raising this important perspective. We agree that the widespread adoption of any agricultural technology depends on its net benefits. We revised the description (Lines 611–620)

    "Continuous green manure intercropping over two years significantly reduced inter-row $CO_2$ emissions, and the HM treatment suppressed fertilization-induced emission peaks. These findings demonstrate that green manure intercropping, particularly mixed legume and non-legume combinations, can effectively alter the spatial pattern of $CO_2$ emissions and mitigate carbon losses. Moreover, this practice provides significant co-benefits like improved soil aeration and fertility, reduced chemical fertilizers, weed suppression, and promoted tea plant growth, thereby offsetting the extra costs in labor and seeds. Therefore, green manure intercropping emerges as a practical and multifunctional strategy for reducing carbon emissions in agroecosystems."

---

## Author Comment (AC3)

**Revision EGUSPHERE-2025-5065**

**Point by point response to the reviewers:**

First of all, we would like to thank the editor and reviewers for the positive and constructive comments. We carefully revised and improved the article. The followings are response to the comments of reviewers. The line number below is indicated based on the **clean version.**

**General Comments**

This study focuses on a subtropical tea plantation and utilizes the static chamber–gas chromatography method to conduct long-term monitoring of $CO_2$ emission fluxes under different green manure intercropping treatments. It systematically reveals the short- and long-term trends and differences in $CO_2$ fluxes across different areas of the tea garden, including between tea rows, and analyzes the impact of key environmental drivers such as soil temperature, moisture, and organic carbon on carbon emissions. The findings provide important insights into the carbon cycling mechanisms in tea plantations under green manure intercropping and hold significant practical value for promoting low-carbon management in tea cultivation. The experimental design is reasonable, the analysis and discussion are thorough, and the structure is clear, making the paper suitable for publication in this journal. The following minor revisions are recommended:

**Response:** Thank you very much for your positive comments. We have revised the manuscript and answered the questions point by point. The line number below is indicated based on the clean version.

**Specific Comments:**

1. Lines 153–154: The baseline physicochemical properties of the soil in the study area are not provided. It is recommended to include initial soil characteristics.

   **Response:** Thank you for the comments. We have supplemented a detailed table of the basic physicochemical properties of the six treatment soils before the experiment (now Table 1) in Sec. 2.1 Lines 164–166.

   "The soil type of the tea plantation is classified as red soil (Ultisol), a typical soil type in this region, and the initial basic physicochemical properties of the six treatment soils are shown in Table 1."

2. Line 171: In addition to the green manure treatment, please specify the type and application rate of fertilizers used.

   **Response:** Thank you for your comment. Regarding the fertilizer treatments mentioned on line 178-180, the specific types and application rates are as follows:

   On October 21, 2022, rapeseed cake was applied as base fertilizer at a rate of 300 kg per mu after trenching and soil turning between the tea rows. On October 28, 2023, base fertilizer consisting of rapeseed cake (65 kg per mu) and compound fertilizer (25 kg per mu) was applied following trenching and soil turning in the tea row inter-rows.

3. Line 185: For consistency, please express time in 24-hour format. Replace "between 9:00 and 11:00 a.m." with "between 09:00 and 11:00 (local time)."

   **Response:** Thank you for the comments. We revised it (Line 192).

4. Line 198: What was the soil sampling depth? This significantly influences soil physicochemical properties. Sampling across multiple soil layers would be more appropriate.

   **Response:** Thank you for the suggestion. The soil sampling depth was 0–20 cm. The core focus of this study is the impact of green manure treatments on soil carbon processes in tea plantations. Since the root systems of green manure and their incorporation primarily concentrate in the 0–20 cm soil layer, sampling at this depth most directly reflects the treatment effects. To avoid destructive deep-layer sampling that could harm the root systems of productive tea plants and to ensure the continuity of long-term observations, this biologically active layer was selected as the representative depth, which is also a commonly adopted standard in comparable studies both domestically and internationally. We recognize the value of stratified sampling and will aim to refine this aspect in future research.

5. Lines 216–217: The method for calculating $CO_2$ fluxes is not clearly described. Please provide details.

   **Response:** Thank you for the comments. The description of the $CO_2$ flux calculation method has been added in the Appendix 1.

6. Lines 308–310: On what basis were the growth stages of green manure defined? Please clarify.

   **Response:** Thank you for the comments. The growth stages of green manure are divided as follows: early growth stage (from mid-November to early April of the following year, primarily functioning for soil surface coverage and water-soil conservation), vigorous growth stage (from mid-April to late May, when biomass and nutrient retention peak), wilting stage (from early June to late July, as the plants naturally wither and prepare for incorporation), and decomposition stage (during August each year, when residues rapidly decompose and release nutrients). This division is based on field phenological observations and biomass dynamics, clarifying the functional transition of each stage within the tea garden ecosystem—from growth and accumulation to incorporation and decomposition—to precisely align with the nutrient management needs of tea garden soils.

7. Lines 498–499: The decrease in cumulative emissions between rows in the second year is attributed to "reduced human disturbance," which is insufficiently supported. Please elaborate with references to relevant literature.

   **Response:** Thank you for your comments. Yes, attributing the decrease in $CO_2$ emissions simply to "reduced anthropogenic disturbance" is overly general. In this study, the cumulative $CO_2$ emissions from the inter-row areas in the second year were lower than those in the first year across all treatments, with a more pronounced reduction observed in the green manure treatments (HM, SM). We believe this change is closely related to the gradual attenuation of the priming effect caused by initial soil disturbance and the continuous improvement in soil structure facilitated by green manure growth. We added more discussion in Sec. 4.2 (Lines 513–523).

"The decrease in emissions can be attributed to the gradual attenuation of the carbon priming effect induced by soil disturbance during the initial experimental phase (Zhou, 2025), coupled with the long-term positive effects of green manure on enhancing soil physical structure and ecosystem stability (Gui et al., 2024). The increase in green manure biomass in the following year indicates that the green manure system is transitioning from an initially disturbed and unstable state toward a more productive and carbon-sequestration-enhanced stable state (Figure A1). This trend not only reflects the improved functioning of the soil ecosystem but also serves as an important driver for further carbon sequestration, contributing significantly to the reduction in inter-row $CO_2$ emissions observed in the following year."

8. Line 541: The discussion on the SOC threshold lacks adequate references. Additional literature should be cited and discussed.

   **Response:** Thank you for the comments. We added more discussion in Sec. 4.3 (Lines 569–574).

   "Studies in different climatic zones of China have revealed that SOC thresholds are influenced by factors such as climate and soil type. In the maritime monsoon climate zone, dual thresholds for $NO_3^-$-N and extractable iron (Fe) have been identified, beyond which their marginal effects on SOC shift significantly. In the continental monsoon climate zone, SOC content increases markedly once a critical threshold of TN is exceeded (Cui, 2025). Additionally, research in alpine ecosystems has shown that SOC components vary along elevation gradients and exhibit distinct thresholds (Zhang, 2025)."

9. Table 1: Please clearly note "Values are mean ± SE" in the table caption or footnote.

   **Response:** Thank you for the comments. We noted "Data shown are means ± SE." in the table 1 footnote.

10. Root biomass data are lacking. The contribution of green manure roots to soil respiration has not been quantified, which may affect the interpretation of $CO_2$ flux sources.

    **Response:** Thank you for your important comment. We have added Figure A1 in Appendix 1, which presents the green manure biomass (including above ground parts and root systems) for the HM and SM treatments. The biomass was measured as fresh weight immediately after sampling and as dry weight after oven-drying at 65°C to constant weight. We added more discussion in Sec. 4.2 (Lines 517–523).

    "The increase in green manure biomass in the following year indicates that the green manure system is transitioning from an initially disturbed and unstable state toward a more productive and carbon-sequestration-enhanced stable state (Figure A1). This trend not only reflects the improved functioning of the soil ecosystem but also serves as an important driver for further carbon sequestration, contributing significantly to the reduction in inter-row $CO_2$ emissions observed in the following year."

11. Within the closed chamber environment, temperature and humidity change over time, potentially influencing $CO_2$ flux measurements. Further analysis on this aspect is recommended.

    **Response:** Thank you for the comments. The static chamber method has an inherent limitation: changes in the chamber micro-environment (temperature, humidity, air pressure) during measurement may affect the calculated $CO_2$ flux. To minimize this impact, the following measures were taken in this experiment:

    1. Controlled measurement duration: The chamber closure time was strictly limited to 21 minutes for each measurement to reduce accumulated deviation in the chamber environment.

    2. Standardized operation: Measurements were conducted during periods of stable weather. A fan was installed inside the chamber to mix the air, while the exterior was wrapped with aluminum foil and sponge to prevent rapid internal temperature rise due to direct sunlight during sampling.

3. Linear relationship verification: We ensured that the $CO_2$ concentration showed a strong linear relationship with time within the selected measurement window ($R^2 > 0.95$), indicating that changes in respiration rate due to chamber environmental variations were not significant during the measurement period.

12. The study does not explore how different green manure treatments regulate microbial activity or the mechanisms by which soil microbial communities drive carbon sequestration. If relevant measurements were not included, this could be addressed in future research.

**Response:** Thank you for the comments. We fully agree that elucidating how green manure treatments drive soil carbon sequestration by regulating microbial activity and community structure is a crucial component for a complete explanation of the ecological mechanisms involved. The current study primarily focuses on the response relationships among green manure, soil physicochemical properties, and $CO_2$ emission fluxes. Due to limitations in research design and duration, it did not include measurements related to microbial community structure and function. Follow-up research will place greater emphasis on investigating the role of soil microorganisms.

13. The language throughout the manuscript should be further refined to avoid repetitive statements, particularly in the Results and Discussion sections.

**Response:** Thank you for the comments. We have refined the language throughout the manuscript to improve conciseness and eliminate repetitive statements. The revisions have focused on consolidating redundant descriptions, employing more precise and varied phrasing, and strengthening the logical flow of the argument.

**Spatially Contrasting CO₂ Dynamics Driven by Green Manure Intercropping in Subtropical Tea Plantations**

Shuo Liu[1,4], Zeping Jin[1,3], Ziyi Chen[1,3], Haolin Li[1,3], Zihan Fan[3], Shaohui Li[3], Haiwang Fu[1,4], Wei He[1], Kunpeng Zang[1], Shuangxi Fang[1,5*], Peng Yan[2]

[revised manuscript text omitted]

In tea rows, the annual mean $CO_2$ fluxes under HMT and SMT treatments were $7.35 \pm 0.44$ mg·m$^{-2}$·min$^{-1}$ and $7.41 \pm 0.45$ mg·m$^{-2}$·min$^{-1}$, respectively, both lower than that of the control (CKT: $8.12 \pm 0.46$ mg·m$^{-2}$·min$^{-1}$) (Fig. 2b). In contrast, in inter-row zones, the annual mean $CO_2$ fluxes were significantly higher under HMG ($9.77 \pm 0.54$ mg·m$^{-2}$·min$^{-1}$) and SMG ($10.83 \pm 0.52$ mg·m$^{-2}$·min$^{-1}$) compared to the control (CKG: $9.07 \pm 0.44$ mg·m$^{-2}$·min$^{-1}$) ($p < 0.05$) (Fig. 2c). Across seasons, CKT generally exhibited higher $CO_2$ fluxes than HMT and SMT, except during winter. In the inter-row zones, both HMG and SMG showed significantly higher fluxes than CKG in summer, while SMG consistently had significantly higher $CO_2$ emissions than both CKG and HMG during the remaining seasons ($p < 0.05$) (Table 12).

Overall, green manure intercropping significantly increased $CO_2$ emissions from inter-rows, but reduced emissions in tea rows. In terms of cumulative annual emissions, HMT and SMT resulted in 3.69 kg·m$^{-2}$ and 3.66 kg·m$^{-2}$ of $CO_2$ emissions, respectively, both lower than the 3.97 kg·m$^{-2}$ under CKT (Fig. 3). Similarly, cumulative $CO_2$ emissions under HMG and SMG remained consistently higher than under CKG, but they declined from 5.76 kg·m$^{-2}$ and 6.43 kg·m$^{-2}$ in the first year to 4.16 kg·m$^{-2}$ and 4.92 kg·m$^{-2}$ in the second year, respectively (Fig. 3). Two consecutive years of green manure intercropping led to a gradual reduction in $CO_2$ emissions from inter-rows, indicating its potential role in long-term emission mitigation in tea plantations. $CO_2$ emissions from inter-rows were substantially higher than those from tea rows. Compared with the control, HM and SM intercropping increased inter-row cumulative $CO_2$ emissions by 12.7% and 28.9%, respectively, while reducing tea-row emissions by 7.1% and 7.9% (Fig. 3a-b). Inter-row zones accounted for 52.6%, 57.3%, and 60.8% of the total annual $CO_2$ emissions in the CK, HM, and SM treatments, respectively (Fig. 3c-d), indicating that the inter-row emissions cannot be ignored.

**3.2 Diurnal $CO_2$ Variations**

[revised manuscript text omitted]
.  The decrease in emissions can be attributed to the gradual attenuation of the carbon priming effect induced by soil disturbance during the initial experimental phase (Zhou, 2025), coupled with the long-term positive effects of green manure on enhancing soil physical structure and ecosystem stability (Gui et al., 2024). The increase in green manure biomass in the following year indicates that the green manure system is transitioning from an initially disturbed and unstable state toward a more productive and carbon-sequestration-enhanced stable state (Figure A1). This trend not only reflects the improved functioning of the soil ecosystem but also serves as an important driver for further carbon sequestration, contributing significantly to the reduction in inter-row $CO_2$ emissions observed in the following year. Therefore, long-term and systematic monitoring of inter-row soil $CO_2$ emissions is essential for accurately assessing the carbon dynamics and mitigation potential of tea plantation ecosystems.

**4.3 Differences in Environmental Drivers**

Soil $CO_2$ fluxes are regulated by multiple environmental factors, including photosynthetic activity or vegetation productivity (Tang et al., 2005), and soil properties such as temperature and moisture (Liu et al., 2023; Widanagamage et al., 2025). Among them, temperature is widely recognized as a primary driver of seasonal variation in soil respiration (Lang et al., 2017). Our results showed that $CO_2$ fluxes in both tea rows and inter-row areas were significantly correlated with soil and air temperatures under different green manure treatments (Fig. 7). In addition, carbon and nitrogen transformation processes driven by microorganisms are closely coupled. Nitrification and denitrification alter $NO_3^-$–N and $NH_4^+$–N levels, thereby influencing soil physicochemical properties and microbial activity. As a result, $CO_2$ emissions exhibit significant positive correlations with nitrogen mineralization, denitrification, and $N_2O$ emissions (Dai et al., 2020). This carbon–nitrogen coupling may interact with the distinctive nutrient uptake characteristics of tea plants, which are ammonium-preferring species with rapid $NH_4^+$ assimilation (Xin et al., 2024). In our study, $CO_2$ fluxes under the CKG treatment were positively correlated with $NH_4^+$–N content (Fig. 9). $NH_4^+$–N levels under this treatment (71.20 mg·kg$^{-1}$) were significantly higher than in the SMG and HMG treatments, whereas the corresponding soil pH value (3.97) was significantly lower ($p < 0.05$). This concurrent high $NH_4^+$–N level and strong acidification is due to ammonium accumulation under conventional fertilization and subsequent $H^+$ release during nitrification (Chen et al., 2021). By contrast, the soil pH under green manure intercropping treatments increased by 0.02–0.11 units compared to the CK treatment (Fig. 6i), suggesting that root exudates and organic matter inputs from green manure buffered soil acidification by reducing $H^+$ release during $NH_4^+$ nitrification. Moreover, the $NO_3^-$–N concentration in the SMG treatment (14.56 mg·kg$^{-1}$) was significantly higher than that in other treatments (Fig. 6e), due to the high biomass of *Vulpia myuros* C., which may reduce nitrate losses via runoff or leaching. Its active root system also improved soil aeration, inhibiting denitrification under anoxic conditions.

SOC content reflects the dynamic balance between organic matter inputs and decomposition (Mo et al., 2024). In our study, SOC levels in the HMT and SMT treatments were higher than those in the CKT treatment, while their cumulative annual $CO_2$ emissions were lower. This indicates that increasing SOC storage can help mitigate greenhouse gas emissions, consistent with findings by Han et al. (2022). However, the HMG and SMG treatments exhibited much higher SOC levels than CKG, while their cumulative $CO_2$ emissions exceeded those of CKG. This implies that once SOC accumulation surpasses a certain threshold, the excess carbon supply may stimulate microbial activity and subsequently increase $CO_2$ emissions (Lim and Choi, 2014). Interestingly, recent studies reveal that SOC thresholds can modulate the impact of nitrogen fertilization on carbon sequestration. In SOC-poor soils, nitrogen inputs tend to promote carbon accumulation and soil aggregation, enhancing SOC storage. Conversely, in SOC-rich soils, nitrogen fertilization may enhance microbial metabolic efficiency and increase microbial residue production (Ling et al., 2025). Studies in different climatic zones of China have revealed that SOC thresholds are influenced by factors such as climate and soil type. In the maritime monsoon climate zone, dual thresholds for $NO_3^-$-N and extractable iron (Fe) have been identified, beyond which their marginal effects on SOC shift significantly. In the continental monsoon climate zone, SOC content increases markedly once a critical threshold of TN is exceeded (Cui, 2025). Additionally, research in alpine ecosystems has shown that SOC components vary along elevation gradients and exhibit distinct thresholds (Zhang, 2025). These insights provide a new perspective for interpreting our results and highlight the importance of identifying threshold values under multifactorial interactions to better assess their effects on $CO_2$ emissions.

[revised manuscript text omitted]


**Tables and Figures**

Table 1. Initial basic physicochemical properties of the six treatments.

| Tpye | pH | $NH_4^+$–N (mg·kg$^{-1}$) | $NO_3^-$–N (mg·kg$^{-1}$) | TN (g·kg$^{-1}$) | TC (g·kg$^{-1}$) | SOC (g·kg$^{-1}$) |
|---|---|---|---|---|---|---|
| CKT | $4.25\pm0.06^a$ | $29.67\pm7.50^b$ | $5.01\pm2.02^b$ | $2.16\pm0.16^{ab}$ | $22.93\pm2.00^a$ | $25.50\pm2.65^a$ |
| HMT | $4.01\pm0.06^{ab}$ | $37.33\pm4.43^{ab}$ | $6.32\pm2.25^b$ | $2.01\pm0.09^{ab}$ | $21.08\pm1.12^a$ | $22.00\pm1.69^a$ |
| SMT | $4.11\pm0.06^{abc}$ | $30.42\pm8.93b$ | $5.10\pm1.88^b$ | $1.96\pm0.12^b$ | $20.47\pm1.12^a$ | $21.83\pm1.92^a$ |
| CKG | $4.16\pm0.02^{ab}$ | $33.20\pm5.90^{ab}$ | $8.59\pm1.46^{ab}$ | $2.41\pm0.19^{ab}$ | $25.70\pm1.99^a$ | $21.68\pm1.82^a$ |
| HMG | $3.98\pm0.05^b$ | $37.90\pm7.98^{ab}$ | $12.73\pm3.94^a$ | $2.70\pm0.35^a$ | $27.77\pm3.57^a$ | $27.98\pm3.55^a$ |
| SMG | $4.00\pm0.03^b$ | $44.40\pm7.33^a$ | $13.36\pm3.82^a$ | $2.31\pm0.28^{ab}$ | $24.92\pm2.86^a$ | $26.88\pm2.53^a$ |

*Data shown are means $\pm$ SE. Different superscript letters indicate the significant difference ($p < 0.05$). CK for control; SM and HM for intercropping types, T for tea row, G for inter-row.

**Table** 2. Seasonal variation in $CO_2$ fluxes from tea rows and inter-row zones under different green manure intercropping treatments.

| Type | Spring (mg·m$^{-2}$·min$^{-1}$) | Summer (mg·m$^{-2}$·min$^{-1}$) | Autumn (mg·m$^{-2}$·min$^{-1}$) | Winter (mg·m$^{-2}$·min$^{-1}$) |
|------|--------|--------|--------|--------|
| CKT | 6.46±0.58$^{bcd}$ | 9.66±0.49$^{a}$ | 9.93±0.68$^{a}$ | 3.60±0.36$^{f}$ |
| HMT | 5.65±0.44$^{def}$ | 7.97±0.39$^{abc}$ | 9.21±0.46$^{a}$ | 4.10±0.43$^{ef}$ |
| SMT | 6.05±0.39$^{cde}$ | 8.99±0.34$^{a}$ | 8.47±0.30$^{ab}$ | 3.71±0.37$^{f}$ |
| CKG | 8.83±1.14$^{cd}$ | 10.63±0.66$^{abc}$ | 9.76±0.83$^{ab}$ | 4.03±0.52$^{e}$ |
| HMG | 9.08±0.49$^{cd}$ | 13.03±0.29$^{a}$ | 9.38±0.39$^{bcd}$ | 4.19±0.29$^{e}$ |
| SMG | 10.76±0.43$^{ab}$ | 12.69±0.73$^{ab}$ | 10.94±0.86$^{abc}$ | 6.03±0.84$^{de}$ |

*Different superscript letters indicate significant differences among treatments and seasons ($p < 0.05$). Data shown are means ± SE. CK for control; SM

and HM for intercropping types, T for tea row, G for inter-row.

[Figure]

(a)

Hang Zhou
Shao Xin
Sheng Zhou

★ City
▲ Station
Altitude(m)

-73
50
          Km (b)

(c)

| Lolium perenne L. || Vulpia myuros C. || Blank Control |
| Trifolium repens L. |

(a)

Hang Zhou
Shao Xin
Sheng Zhou

★ City
▲ Station
Altitude(m)

-73
50
          Km (b)

0.8m          1.25m
        1m (c)

| Lolium perenne L. || Vulpia myuros C. || Blank Control |
| Trifolium repens L. |

[Figure]

**Figure 1.** (a) Geographic location of the study area in Shengzhou City, Zhejiang

Province, China; (b) field layout of the tea plantation experiment; (c) photos of *Lolium*

*perenne* L. and *Trifolium repen*s L. plot, and *Vulpia myuros* C. plot, the blankare control plot, respectively.

[Figure]

**Figure 2.** Dynamics of (a) air temperature and precipitation, (b) CO₂ fluxes from tea rows, and (c) CO₂ fluxes from inter-rows during the observation period (2022–2024). Black, green and orange arrows represent the timings of fertilization, grass planting and tea pruning, respectively. Flux data are presented as mean ± SE. CK for control; SM

and HM for intercropping types, T for tea row, G for inter-row.

[Figure]

**Figure 3.** (a, b) Annual cumulative $CO_2$ emissions from tea rows and inter-rows under different green manure intercropping treatments; (c, d) contribution of tea rows and inter-rows to total annual $CO_2$ emissions under each treatment. Data shown are means ± SE. Different superscript letters denote statistically significant differences ($p < 0.05$). CK for control; SM and HM for intercropping types, T for tea row, G for inter-row.

[Figure]

**Figure 4.** Diurnal variation in $CO_2$ fluxes from (a, c, e, g) tea rows and (b, d, f, h) inter-row zones under different green manure intercropping treatments across seasons: (a–b) spring, (c–d) summer, (e–f) autumn, and (g–h) winter. CK for control; SM and HM for intercropping types, T for tea row, G for inter-row.

[Figure]

**Figure 5.** Temporal dynamics of $CO_2$ fluxes under green manure (a, b) growth stages and (c, d) management events in tea plantations. Growth stages include: early growth (mid-November to early April), vigorous growth (mid-April to late May), wilting (early June to late July), and decomposition (August). CK for control; SM and HM for intercropping types, T for tea row, G for inter-row.

[Figure]

**Figure 6.** Basic physicochemical properties of soil in tea rows and inter-rows under different green manure intercropping treatments. CK for control; SM and HM for intercropping types, T for tea row, G for inter-row.

[Figure]

**Figure 7.** Pairwise correlations between environmental factors and their relationships with CO$_2$ fluxes under different green manure treatments (*$p < 0.05$, **$p < 0.01$, ***$p$

$< 0.001$). CK for control; SM and HM for intercropping types, T for tea row, G for interrow.

[Figure]

**Figure 8.** Canonical correspondence analysis (CCA) showing the influence of soil physicochemical properties on $CO_2$ emissions from tea rows and inter-rows under different green manure treatments. CK for control; SM and HM for intercropping types,

T for tea row, G for inter-row.

**Appendix 1**

The $CO_2$ flux was calculated using the following equation:

$$F = \rho_0 \times \frac{P}{P_O} \times \frac{T_O}{T+T_O} \times \frac{V}{M} \times \frac{\Delta c}{\Delta t} \hspace{2cm} (1)$$

where F is the $CO_2$ flux (mg·m$^{-2}$·min$^{-1}$); $\rho_0$ is the density of $CO_2$ under standard conditions (1.98 kg·m$^{-3}$); $P_o$ and $T_o$ are the standard atmospheric pressure (101.325 kPa)

and temperature (273.15 K), respectively; P and T are the atmospheric pressure (kPa)

and absolute temperature (K) at the time of sampling; V and M are the volume (m$^3$) and bottom area (m$^2$) of the chamber, respectively; $\triangle c/\triangle t$ is the slope of the linear or nonlinear regression of $CO_2$ concentration over time.

First, the raw $CO_2$ concentration readings were calibrated with standard gases.

Then, linear regression was performed to fit their rate of change over time. Finally, $CO_2$

flux was calculated using the standard flux formula by incorporating the chamber volume, base area, and the measured atmospheric pressure and temperature.

[Figure]

**Figure A1.** Temporal trend of green manure indexes

**Spatially Contrasting CO₂ Dynamics Driven by Green Manure Intercropping in Subtropical Tea Plantations**

Shuo Liu[1,4], Zeping Jin[1,3], Ziyi Chen[1,3], Haolin Li[1,3], Zihan Fan[3], Shaohui Li[3], Haiwang Fu[1,4], Wei He[1], Kunpeng Zang[1], Shuangxi Fang[1,5*], Peng Yan[2]

[1] Zhejiang Carbon Neutral Innovation Institute & Zhejiang International Cooperation Base for Science and Technology on Carbon Emission Reduction and Monitoring, Zhejiang University of Technology, Hangzhou 310014, China

[2] Key Laboratory of Tea Quality and Safety Control, Ministry of Agriculture, Tea Research Institute, Chinese Academy of Agricultural Sciences, Hangzhou 310008, China

[3] College of Environment, Zhejiang University of Technology, Hangzhou 310014, China

[4] Shaoxing Research Institute, Zhejiang University of Technology, Shaoxing 312077, China

[5] State Key Laboratory of Green Chemical Synthesis and Conversion, Zhejiang University of Technology, Hangzhou 310014, China

Correspondence authors:

Shuangxi Fang, E-mail: fangsx@zjut.edu.cn

**Abstract:**

Tea plantations are important contributors to greenhouse gas emissions due to intensive fertilization and continuous cultivation. However, the mechanisms by which green manure intercropping regulates soil $CO_2$ dynamics in these systems remain poorly understood. We employed the static chamber method over a two-year period, with sampling conducted weekly, to investigate how intercropping with *Vulpia myuros* (SM) and a legume–nonlegume mixture of *Lolium perenne* and *Trifolium repens* (HM) influenced spatial $CO_2$ flux dynamics compared with a no-intercropping control (CK) from tea rows and inter-row zones in a subtropical tea plantation. Distinct seasonal variations were observed, with $CO_2$ fluxes peaking in summer and autumn and declining in spring and winter. Average tea-row fluxes were 8.7% and 9.5% lower under SM and HM, respectively, compared to CK, indicating emission reductions with intercropping. In contrast, average inter-row fluxes increased by 19.4% under SM and 7.7% under HM, demonstrating pronounced spatial contrasts. Diurnal patterns generally exhibited midday peaks (12:00–14:00), especially in summer and autumn across all tea-rows, and short-term $CO_2$ 
[revised manuscript text omitted]

In tea rows, the annual mean $CO_2$ fluxes under HMT and SMT treatments were

$7.35 \pm 0.44$ mg·m$^{-2}$·min$^{-1}$ and $7.41 \pm 0.45$ mg·m$^{-2}$·min$^{-1}$, respectively, both lower than that of the control (CKT: $8.12 \pm 0.46$ mg·m$^{-2}$·min$^{-1}$) (Fig. 2b). In contrast, in inter-row zones, the annual mean $CO_2$ fluxes were significantly higher under HMG ($9.77 \pm 0.54$

mg·m$^{-2}$·min$^{-1}$) and SMG ($10.83 \pm 0.52$ mg·m$^{-2}$·min$^{-1}$) compared to the control (CKG:

$9.07 \pm 0.44$ mg·m$^{-2}$·min$^{-1}$) ($p < 0.05$) (Fig. 2c). Across seasons, CKT generally exhibited higher $CO_2$ fluxes than HMT and SMT, except during winter. In the inter-row zones, both HMG and SMG showed significantly higher fluxes than CKG in summer, while SMG consistently had significantly higher $CO_2$ emissions than both CKG and

HMG during the remaining seasons ($p < 0.05$) (Table 2).

Overall, green manure intercropping significantly increased $CO_2$ emissions from inter-rows, but reduced emissions in tea rows. In terms of cumulative annual emissions,

HMT and SMT resulted in 3.69 kg·m$^{-2}$ and 3.66 kg·m$^{-2}$ of $CO_2$ emissions, respectively, both lower than the 3.97 kg·m$^{-2}$ under CKT (Fig. 3). Similarly, cumulative $CO_2$

emissions under HMG and SMG remained consistently higher than under CKG, but they declined from 5.76 kg·m$^{-2}$ and 6.43 kg·m$^{-2}$ in the first year to 4.16 kg·m$^{-2}$ and 4.92

kg·m$^{-2}$ in the second year, respectively (Fig. 3). Two consecutive years of green manure intercropping led to a gradual reduction in $CO_2$ emissions from inter-rows, indicating its potential role in long-term emission mitigation in tea plantations. $CO_2$ emissions from inter-rows were substantially higher than those from tea rows. Compared with the control, HM and SM intercropping increased inter-row cumulative $CO_2$ emissions by

12.7% and 28.9%, respectively, while reducing tea-row emissions by 7.1% and 7.9%

(Fig. 3a-b). Inter-row zones accounted for 52.6%, 57.3%, and 60.8% of the total annual

$CO_2$ emissions in the CK, HM, and SM treatments, respectively (Fig. 3c-d), indicating that the inter-row emissions cannot be ignored.

**3.2 Diurnal $CO_2$ Variations**

$CO_2$ fluxes in the tea plantation exhibited pronounced diurnal variations across all seasons, particularly in spring and summer (Fig. 4), likely influenced by the growth stages of green manure species. In spring, $CO_2$ fluxes in tea rows under all treatments showed a similar diurnal trend: an initial decline followed by a rapid increase. HMT

and SMT reached their minimum fluxes at 08:00 (local time), with values of $-3.74$

$mg \cdot m^{-2} \cdot min^{-1}$ and $-3.80$ $mg \cdot m^{-2} \cdot min^{-1}$, respectively, then rose sharply and stabilized in the afternoon. The diurnal amplitudes under HMT and SMT were notably greater than that of the control (CKT) (Fig. 4a). In the inter-row zones, the diurnal patterns under green manure treatments differed notably from the control (Fig. 4b). CKG displayed a unimodal pattern with a peak at 12:00 (12.74 $mg \cdot m^{-2} \cdot min^{-1}$) and a trough at 08:00 (5.45

$mg \cdot m^{-2} \cdot min^{-1}$), resulting in an amplitude of 7.29 $mg \cdot m^{-2} \cdot min^{-1}$. In contrast, HMG and

SMG exhibited later peaks at 16:00 (23.26 $mg \cdot m^{-2} \cdot min^{-1}$) and 14:00 (24.17

$mg \cdot m^{-2} \cdot min^{-1}$), respectively, with troughs also at 08:00 (HMG: 12.28 $mg \cdot m^{-2} \cdot min^{-1}$;

SMG: 12.43 $mg \cdot m^{-2} \cdot min^{-1}$). Both treatments showed substantially higher amplitudes than CKG.

Summer exhibited the most pronounced diurnal variation of $CO_2$ fluxes across all seasons. In tea rows, CKT, HMT, and SMT followed a bimodal pattern, with peaks at

02:00 and 12:00, and a trough at 08:00. Their respective diurnal amplitudes were 12.96,

6.70, and 10.10 $mg \cdot m^{-2} \cdot min^{-1}$ (Fig. 4c). In the inter-rows, the amplitudes were relatively lower, 7.72, 8.12, and 7.79 $mg \cdot m^{-2} \cdot min^{-1}$ for CKG, HMG, and SMG, respectively, indicating smaller fluctuations compared to tea rows (Fig. 4d). Notably, summer also showed the most distinct contrast between tea rows and inter-rows: CKT

recorded the highest average flux in the tea rows, while CKG had the lowest in the interrows.

In autumn, tea-row fluxes under all treatments exhibited a unimodal pattern, with minima at 08:00 and peaks at 14:00. The diurnal amplitudes were 11.27, 8.02, and 12.75

$mg \cdot m^{-2} \cdot min^{-1}$ for CKT, HMT, and SMT, respectively (Fig. 4e). In the inter-rows, HMG

and SMG displayed relatively stable diurnal trends, whereas CKG showed a bimodal pattern with peaks at 06:00 and 16:00, and a greater amplitude than both HMG and

SMG (Fig. 4f).

In winter, $CO_2$ fluxes showed the most stable diurnal variation of the year. In tea rows, amplitudes were only 2.96, 2.84, and 4.92 $mg \cdot m^{-2} \cdot min^{-1}$ for CKT, HMT, and

[revised manuscript text omitted]

Conversely, in SOC-rich soils, nitrogen fertilization may enhance microbial metabolic efficiency and increase microbial residue production (Ling et al., 2025). Studies in different climatic zones of China have revealed that SOC thresholds are influenced by factors such as climate and soil type. In the maritime monsoon climate zone, dual thresholds for $NO_3^-$-N and extractable iron (Fe) have been identified, beyond which their marginal effects on SOC shift significantly. In the continental monsoon climate zone, SOC content increases markedly once a critical threshold of TN is exceeded (Cui,

2025). Additionally, research in alpine ecosystems has shown that SOC components vary along elevation gradients and exhibit distinct thresholds (Zhang, 2025). These insights provide a new perspective for interpreting our results and highlight the importance of identifying threshold values under multifactorial interactions to better assess their effects on $CO_2$ emissions.

[revised manuscript text omitted]

 **Tables and Figures**

 **Table 1.** Initial basic physicochemical properties of the six treatments.

| Tpye | pH | $NH_4^+$–N (mg·kg$^{-1}$) | $NO_3^-$–N (mg·kg$^{-1}$) | TN (g·kg$^{-1}$) | TC (g·kg$^{-1}$) | SOC (g·kg$^{-1}$) |
|---|---|---|---|---|---|---|
| CKT | 4.25±0.06[a] | 29.67±7.50[b] | 5.01±2.02[b] | 2.16±0.16[ab] | 22.93±2.00[a] | 25.50±2.65[a] |
| HMT | 4.01±0.06[ab] | 37.33±4.43[ab] | 6.32±2.25[b] | 2.01±0.09[ab] | 21.08±1.12[a] | 22.00±1.69[a] |
| SMT | 4.11±0.06[abc] | 30.42±8.93b | 5.10±1.88[b] | 1.96±0.12[b] | 20.47±1.12[a] | 21.83±1.92[a] |
| CKG | 4.16±0.02[ab] | 33.20±5.90[ab] | 8.59±1.46[ab] | 2.41±0.19[ab] | 25.70±1.99[a] | 21.68±1.82[a] |
| HMG | 3.98±0.05[b] | 37.90±7.98[ab] | 12.73±3.94[a] | 2.70±0.35[a] | 27.77±3.57[a] | 27.98±3.55[a] |
| SMG | 4.00±0.03[b] | 44.40±7.33[a] | 13.36±3.82[a] | 2.31±0.28[ab] | 24.92±2.86[a] | 26.88±2.53[a] |

 *Data shown are means $\pm$ SE. Different superscript letters indicate the significant
 difference ($p < 0.05$). CK for control; SM and HM for intercropping types, T for tea
 row, G for inter-row.

**Table 2.** Seasonal variation in $CO_2$ fluxes from tea rows and inter-row zones under different green manure intercropping treatments.

| Type | Spring (mg·m$^{-2}$·min$^{-1}$) | Summer (mg·m$^{-2}$·min$^{-1}$) | Autumn (mg·m$^{-2}$·min$^{-1}$) | Winter (mg·m$^{-2}$·min$^{-1}$) |
|------|------|------|------|------|
| CKT | 6.46±0.58[bcd] | 9.66±0.49[a] | 9.93±0.68[a] | 3.60±0.36[f] |
| HMT | 5.65±0.44[def] | 7.97±0.39[abc] | 9.21±0.46[a] | 4.10±0.43[ef] |
| SMT | 6.05±0.39[cde] | 8.99±0.34[a] | 8.47±0.30[ab] | 3.71±0.37[f] |
| CKG | 8.83±1.14[cd] | 10.63±0.66[abc] | 9.76±0.83[ab] | 4.03±0.52[e] |
| HMG | 9.08±0.49[cd] | 13.03±0.29[a] | 9.38±0.39[bcd] | 4.19±0.29[e] |
| SMG | 10.76±0.43[ab] | 12.69±0.73[ab] | 10.94±0.86[abc] | 6.03±0.84[de] |

*Different superscript letters indicate significant differences among treatments and seasons ($p < 0.05$). Data shown are means ± SE. CK for control; SM and HM for intercropping types, T for tea row, G for inter-row.

[Figure]

**Figure 1.** (a) Geographic location of the study area in Shengzhou City, Zhejiang

Province, China; (b) field layout of the tea plantation experiment; (c) photos of *Lolium*

*perenne* L. and *Trifolium repen*s L. plot, and *Vulpia myuros* C. plot, the blank control plot, respectively.

[Figure]

**Figure 2.** Dynamics of (a) air temperature and precipitation, (b) CO₂ fluxes from tea rows, and (c) CO₂ fluxes from inter-rows during the observation period (2022–2024).

Black, green and orange arrows represent the timings of fertilization, grass planting and tea pruning, respectively. Flux data are presented as mean ± SE. CK for control; SM

and HM for intercropping types, T for tea row, G for inter-row.

[Figure]

**Figure 3.** (a, b) Annual cumulative $CO_2$ emissions from tea rows and inter-rows under different green manure intercropping treatments; (c, d) contribution of tea rows and inter-rows to total annual $CO_2$ emissions under each treatment. Data shown are means $\pm$ SE. Different superscript letters denote statistically significant differences ($p < 0.05$). CK for control; SM and HM for intercropping types, T for tea row, G for inter-row.

[Figure]

**Figure 4.** Diurnal variation in $CO_2$ fluxes from (a, c, e, g) tea rows and (b, d, f, h) interrow zones under different green manure intercropping treatments across seasons: (a–b)

spring, (c–d) summer, (e–f) autumn, and (g–h) winter. CK for control; SM and HM for intercropping types, T for tea row, G for inter-row.

[Figure]

**Figure 5.** Temporal dynamics of $CO_2$ fluxes under green manure (a, b) growth stages
and (c, d) management events in tea plantations. Growth stages include: early growth
(mid-November to early April), vigorous growth (mid-April to late May), wilting (early
June to late July), and decomposition (August). CK for control; SM and HM for
intercropping types, T for tea row, G for inter-row.

[Figure]

**Figure 6.** Basic physicochemical properties of soil in tea rows and inter-rows under different green manure intercropping treatments. CK for control; SM and HM for intercropping types, T for tea row, G for inter-row.

[Figure]

**Figure 7.** Pairwise correlations between environmental factors and their relationships with CO₂ fluxes under different green manure treatments (*$p < 0.05$, **$p < 0.01$, ***$p$

$< 0.001$). CK for control; SM and HM for intercropping types, T for tea row, G for interrow.

[Figure]

**Figure 8.** Canonical correspondence analysis (CCA) showing the influence of soil physicochemical properties on $CO_2$ emissions from tea rows and inter-rows under different green manure treatments. CK for control; SM and HM for intercropping types,

T for tea row, G for inter-row.

**Appendix 1**

The $CO_2$ flux was calculated using the following equation:

$$F = \rho_0 \times \frac{P}{P_O} \times \frac{T_O}{T+T_O} \times \frac{V}{M} \times \frac{\Delta c}{\Delta t} \tag{1}$$

where F is the $CO_2$ flux ($mg \cdot m^{-2} \cdot min^{-1}$); $\rho_0$ is the density of $CO_2$ under standard conditions ($1.98\ kg \cdot m^{-3}$); $P_O$ and $T_O$ are the standard atmospheric pressure (101.325 kPa) and temperature (273.15 K), respectively; $P$ and $T$ are the atmospheric pressure (kPa) and absolute temperature (K) at the time of sampling; $V$ and $M$ are the volume ($m^3$) and bottom area ($m^2$) of the chamber, respectively; $\Delta c/\Delta t$ is the slope of the linear or nonlinear regression of $CO_2$ concentration over time.

First, the raw $CO_2$ concentration readings were calibrated with standard gases. Then, linear regression was performed to fit their rate of change over time. Finally, $CO_2$ flux was calculated using the standard flux formula by incorporating the chamber volume, base area, and the measured atmospheric pressure and temperature.

[Figure]

**Figure A1.** Temporal trend of green manure indexes